# Outcome Rewards Do Not Guarantee Verifiable or Causally Important Reasoning

Qinan Yu [1]  Alexa Tartaglini [1]  Peter Hase [1 2]  Carlos Guestrin [1]  Christopher Potts [1]

## Abstract

Reinforcement Learning from Verifiable Rewards (RLVR) on chain-of-thought reasoning has become a standard part of language model post-training recipes. A common assumption is that the reasoning chains trained through RLVR reliably represent how a model gets to its answer. In this paper, we develop two metrics for critically examining this assumption: *Causal Importance of Reasoning* (CIR), which measures the cumulative effect of reasoning tokens on the final answer, and *Sufficiency of Reasoning* (SR), which measures whether a verifier can arrive at an unambiguous answer based on the reasoning alone. Through experiments with the Qwen2.5 model series and ReasoningGym tasks, we find that: (1) While RLVR does improve task accuracy, it does not reliably improve CIR or SR, calling the role of reasoning in model performance into question. (2) A small amount of SFT before RLVR can be a remedy for low CIR and SR. (3) CIR and SR can be improved even without SFT by applying auxiliary CIR/SR rewards on top of the outcome-based reward. This joint reward matches the accuracy of RLVR while also leading to causally important and sufficient reasoning. These results show that RLVR does not always lead models to rely on reasoning in the way that is commonly thought, but this issue can be remedied with simple modifications to the post-training procedure. Our code is available at https://github.com/yuqinan/cir-sr/.

## 1. Introduction

Reinforcement learning from verifiable rewards (RLVR) has become an important part of post-training efforts to produce models that *think before they answer* (Guo et al., 2025; Lambert et al., 2025), typically in domains where correctness can be automatically verified. Since the reward in RLVR is determined only by the final outcome (answer), and not the rest of the model output (reasoning), this reward is known as an outcome-based reward.

Beyond improving task performance, training models to emit reasoning chains opens up the opportunity for validating and monitoring the model's reasoning process (Hase et al., 2020; Korbak et al., 2025). However, the ability to interpret a model's chain-of-thought rests on two assumptions: (1) that the chain-of-thought is *causally important*, i.e., it causally reflects the reasoning process the model used to arrive at its answer (Jacovi & Goldberg, 2020); and (2) that the chain-of-thought is *verifiable*, i.e., the important reasoning steps are explicit and reliable enough to lead an external verifier to the same answer (Kirchner et al., 2024). A model could produce detailed reasoning chains that are easily checked for correctness by a third party but which are not causally linked to its final answer (verifiable but not causally important), or its final answer might strongly depend on a series of reasoning steps that are vague or leave out key steps (causally important but not verifiable). In either case, the reasoning chain does not provide enough reliable information for an external reader to validate the model's solution process. If we hope to analyze how models solve tasks, it is important to understand when reasoning chains possess (or lack) these two desirable properties and how post-training recipes affect them.

We study the causal importance and verifiability of the reasoning chains produced by RLVR by introducing two complementary metrics (Section 3). *Causal Importance of Reasoning* (CIR) measures how much a model's predicted answer distribution depends on the associated reasoning trace by comparing model predictions conditioned on progressively truncated versions of the reasoning chain. *Sufficiency of Reasoning* (SR) measures how self-contained and verifiable a reasoning trace is by evaluating whether the reasoning points unambiguously to a particular answer choice, which is measured by asking a verifier model to predict the answer based on the reasoning. Specifically, we measure whether the verifier's guess changes when given the *question* and the

---

[1]Stanford University [2]Schmidt Sciences. Correspondence to: Qinan Yu <qinanyu@stanford.edu>.

*Proceedings of the $43^{rd}$ International Conference on Machine Learning*, Seoul, South Korea. PMLR 306, 2026. Copyright 2026 by the author(s).

prompt, alongside the reasoning, compared to when only the reasoning is provided. This shows whether the reasoning was sufficient on its own for determining an answer.

Employing these two metrics, we analyze how causal importance and verifiability of reasoning chains evolve throughout the RLVR training process. Our experiments focus on the Qwen2.5 series (Qwen et al., 2025), using instruction-tuned models of varying sizes (1.5B, 3B, 7B) as our starting point. We also run additional experiments on Llama3.2-3B (Grattafiori et al., 2024) to test across different model families. We apply RLVR training to these models on a range of tasks to address the following research questions:

**R1** Does RLVR improve the causal importance and verifiability of the reasoning chains?

**R2** How does supervised learning before RLVR affect reasoning causal importance and verifiability?

**R3** Can we jointly improve accuracy, causal importance, and verifiability during RLVR?

On R1: In Section 5, our results show that RLVR does not reliably translate into causally important or verifiable reasoning: across 40 tasks in ReasoningGym (Stojanovski et al., 2025), 19 of the tasks show a decline in CIR and 17 of them in SR. We find that only the subset of tasks where accuracy goes up more than 50 points during RLVR shows reliably increased CIR and SR. We hypothesize that this is because models solve these tasks in ReasoningGym without needing reasoning at all. To validate this, we compare running RLVR with and without reasoning chains. We find that on tasks where the model's reasoning chains have low CIR and SR at the end of training, training without reasoning yields similar accuracy improvements.

On R2: In Section 6, we show that supervised learning before RLVR with a small set of high quality reasoning traces substantially boosts both CIR and SR. In fact, when we follow this supervised learning with RLVR (known as SL-before-RL), we see that CIR and SR improve during RLVR.

On R3: In Section 7, we augment RLVR with explicit CIR/SR-based rewards that directly target these properties. We find that this joint reward improves CIR and SR while preserving final accuracy, without using the expert reasoning traces needed for supervised learning.

Taken together, our results suggest that outcome-based rewards alone are insufficient to ensure causally important and verifiable reasoning, and that designing training objectives and evaluation metrics that directly target these properties is crucial for building trustworthy and reliable reasoning models.

## 2. Related Work

**Training Models to Reason with RLVR**    Language models are increasingly trained to produce explicit reasoning traces, rather than only final answers (Guo et al., 2025; OpenAI et al., 2024). These models are typically trained to generate reasoning and rewarded for final-answer correctness (Wen et al., 2025; Lambert et al., 2025). RLVR is popular in verifiable domains such as math (Team et al., 2025; Shao et al., 2024; 2025) and code (Da et al., 2025; Jin et al., 2025). Shao et al. (2024) proposed GRPO for improved efficiency, and subsequent work introduced variants and extensions (Yu et al., 2025; Zheng et al., 2025).

**Faithful Reasoning**    *Faithfulness* is central for trustworthy and interpretable reasoning (Lyu et al., 2023). Here, we focus on chain-of-thought reasoning (Wei et al., 2023), although any textual explanation from LLMs can be evaluated for faithfulness, including post-hoc rationales (Chen et al., 2023). Faithfulness asks whether the emitted chain-of-thought accurately reflects why the model produced its answer. The causal relationship between the reasoning chain and the final answer is one dimension used to evaluate faithfulness (Lanham et al., 2023). Following a causal perspective (Pearl, 2000), Lyu et al. (2023) and Lanham et al. (2023) operationalize this notion by perturbing parts of a trace and measuring changes in the output. Subsequent work evaluates trace faithfulness across settings and shows that traces can be unfaithful—for example, omitting key information that drives the final answer (Shen et al., 2025; Arcuschin et al., 2025; Matton et al., 2025; Xiong et al., 2025). Other work proposes supervised training objectives for improving faithfulness (Paul et al., 2024) and studies how common post-training procedures influence faithfulness (McMillan et al., 2025; Chen et al., 2025). In this paper, we propose novel rewards that improve causal importance and verifiability during RLVR, using open-source models. We note that our Causal Importance of Reasoning (CIR) metric is similar to the TRACE metric in concurrent work (Wang et al., 2025), where it is used for detecting reward hacking in math and code tasks.

**Verifiable Reasoning**    *Verifiability* asks whether a reasoning trace can be independently checked step-by-step, enabling monitoring for misalignment (Chan et al., 2025; Guan et al., 2025). Benchmarking efforts such as Prasad et al. 2023 and Jacovi et al. 2024 provide fine-grained annotations of reasoning chains to evaluate automatic verifiers and reveal that small local errors can invalidate an otherwise plausible chain. Neuro-symbolic approaches translate reasoning steps into formal statements and apply logic-based consistency tests, enabling automated detection of unsupported or inconsistent steps (Feng et al., 2025). Relatedly, environments that provide *verifiable rewards* support eval-

## (a) **CIR**: **C**ausal **I**mportance of **R**easoning

Question: "..."

*thinking tokens*

<think> $t_1$ $t_2$ $\cdots$ $t_T$ </think> ⭐ answer

*truncate the CoT at each token $t_k$ ($k \in [1, T]$)*

*for each $k$: is the final* ⭐*answer equally likely given the truncated CoT?*

✂

$P(\bigstar | t_{1:k})$

$\overset{?}{=} P(\bigstar | t_{1:T})$

$\mathbf{CIR} = \begin{cases} \text{low} & \bigstar \text{ is already likely at } k \ll T \\ \text{high} & \bigstar \text{ is only likely at } k \approx T \end{cases}$

## (b) **SR**: **S**ufficiency of **R**easoning

Question: "..."

<think> $\cdots$ </think> answer

*verifier*

thinking

question + thinking

$y_1$

$y_2$

*verifier answers*

$\mathbf{SR} = \begin{cases} 1 & y_1 = y_2 \\ 0 & y_1 \neq y_2 \end{cases}$

*check: does seeing the question change the answer?*

*Figure 1.* Overview of our two metrics. (a) Causal Importance of Reasoning (CIR). (b) Sufficiency of Reasoning (SR).

uation and training loops where correctness is externally checkable rather than inferred from surface plausibility (Stojanovski et al., 2025). Our Sufficiency of Reasoning metric is inspired by work on reasoning legibility (Kirchner et al., 2024). While Kirchner et al. (2024) also train models to produce more verifiable reasoning, we show how to jointly optimize for accuracy, faithfulness, and verifiability during RLVR using open-source models.

## 3. Metrics

A high-quality reasoning chain should be both *causally important* and *informationally sufficient* for determining the final answer. From a dependency perspective, this means that the answer should depend on the reasoning chain rather than solely on the question, and that the reasoning chain should be unambiguous without further reliance on the question. We formalize these complementary requirements using conditional independence and introduce two metrics to evaluate reasoning quality: *Causal Importance of Reasoning (CIR)* and *Sufficiency of Reasoning (SR)*. CIR evaluates whether the answer $y$ is independent of the reasoning chain $t$ given the question $q$, capturing whether the reasoning is actually used by the model. In contrast, SR measures whether the answer $y$ is independent of the question $q$ given the reasoning chain $t$, capturing whether the reasoning is clear and complete. For a high-quality reasoning chain, $y$ should depend on $t$ rather than solely on $q$, while $t$ should be informative enough that conditioning on $q$ provides little additional information about $y$.

### 3.1. Causal Importance of Reasoning (CIR)

For question $q$ with a reasoning chain $t = (t_1, \ldots, t_T)$ and answer $y$, we define truncation at prefix length $k$ as keeping only the first $k$ tokens in the reasoning chain, i.e. $t_{1:k} = (t_1, \ldots, t_k)$ and "forcing" the model to answer the question

early. Let $p_k = p_\phi(y \mid q, t_{1:k})$ and $p_T = p_\phi(y \mid q, t_{1:T})$ where $\phi$ is the reasoning model.

To obtain a divergence metric between $p_k$ and $p_T$, we treat these scalars as parameters for Bernoulli distributions and compute the Jensen-Shannon divergence between them, $\text{JS}\big(\text{Bernoulli}(p_k)||\text{Bernoulli}(p_T)\big)$. Then, we compute our final CIR score by averaging the JS scores across token indices:

$$\text{CIR} = \frac{1}{T} \sum_{k=1}^{T} \text{JS}\big(\text{Bernoulli}(p_k)||\text{Bernoulli}(p_T)\big). \quad (1)$$

If the reasoning chains are causally used, then truncating the trace to a shorter prefix $t_{1:k}$ will significantly change the model's prediction of the final answer. Formally, if the CIR is 0, $y \perp\!\!\!\perp t \mid q$; the answer $y$ is solely dependent on the question $q$ rather than the reasoning chain $t$.

### 3.2. Sufficiency of Reasoning (SR)

Given a question token sequence $q$, a reasoning chain $t$, and a generated answer sequence $y$ from the model, we first sanitize the reasoning chain to prevent trivial answer leakage by removing any explicit answer content through exact match:

$$t' = \text{RemoveAnswer}(t, y). \quad (2)$$

We then instruct a verifier $\theta$ to predict an answer distribution from the sanitized reasoning chain, either (i) conditioned on the original question or (ii) without access to it (see Appendix B for the prompts used to create these conditions). If the reasoning is sufficient, conditioning on $q$ should add little information, so the probability-level predictions should be nearly unchanged:

$$p_\theta(\cdot \mid q, t') \approx p_\theta(\cdot \mid t'). \quad (3)$$

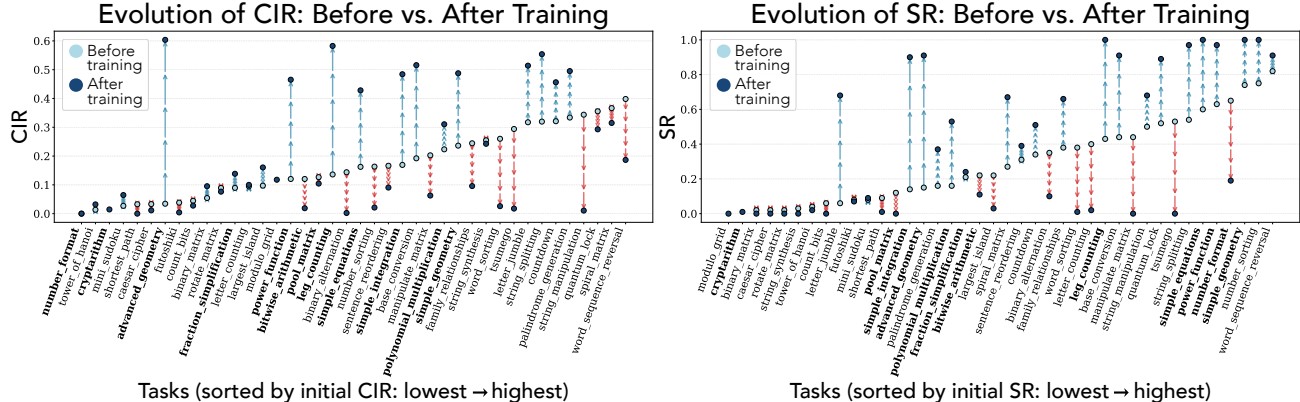

*Figure 2.* **Causal Importance of Reasoning (CIR) and Sufficiency of Reasoning (SR) before vs. after RLVR, for Qwen 2.5-3B.** Left: task-level CIR, sorted by initial CIR. Right: task-level SR, sorted by initial SR. Some tasks (e.g. `countdown`, `advanced_geometry`, `power_function`) improve CIR after training, but many tasks see CIR and/or SR decrease toward 0, and tasks that start with a low CIR and SR rarely improve in these metrics through RLVR. The bolded tasks are in the math domain.

where $p_\theta$ denotes the verifier's probability-level prediction.

The verifier model should be a very strong model capable of making use of sophisticated reasoning patterns. In our main experiments, we rely on gpt-4o-mini. We show that SR is essentially independent of the choice of strong verifier (see Appendix G). Because such models are generally accessible only via APIs that do not provide direct access to their full predicted output distributions, we approximate SR as agreement between the verifier's decoded outputs under the two conditioning settings:

$$\text{SR}(q,t) = \begin{cases} 1 & \text{if } \hat{y}(q,t') = \hat{y}(t'), \\ 0 & \text{otherwise.} \end{cases} \quad (4)$$

where $\hat{y}(q,t')$ is the decoded answer from $p_\theta(\cdot \mid q, t')$. (In our experiments, this is based on greedy decoding.)

As SR approaches 1, $q$ provides no information about the final answer $y$ beyond what is encoded in $t$. However, we emphasize that SR is a permissive measure. Since we evaluate only whether the predicted and true final answers match, a reasoning chain can achieve high SR even without containing every necessary reasoning step. On the other hand, if SR is low, then we can be confident that the reasoning chain is vague and unclear, providing hardly any verifiable clues.

## 4. Setup

**Datasets** To evaluate reasoning quality under RLVR, we rely on ReasoningGym (Stojanovski et al., 2025). ReasoningGym provides a diverse range of tasks, including tasks that are outside of the very popular areas of Math and Code (Guha et al., 2025). This makes ReasoningGym an especially valuable benchmark for isolating the effects of RLVR on reasoning chains. Every task is evaluated with a specific rule-based function on the answer that returns a score be-

tween 0 and 1. To manage our computational constraints, we select 40 of the over 100 ReasoningGym tasks from diverse categories. First, we choose tasks where the base model achieves intermediate accuracy (neither perfect nor consistently zero), allowing the model to meaningfully learn the task. For example, we omit the 'time interval' task, for which Qwen 2.5-3B already achieves 0.80 accuracy with pass@1. Second, we prioritize multi-class tasks over binary ones. Our full list of selected tasks is in Appendix A.

Beyond the ReasoningGym domain, we run additional experiments on Math-Hard (Hendrycks et al., 2021) and include the additional results in Appendix I.

**Models** We begin with instruction-tuned models that have not undergone RLVR training to study how RLVR affects their reasoning traces. Specifically, we use Qwen2.5-1.5B, 3B, 7B (Qwen et al., 2025) and Llama3.2-3B (Grattafiori et al., 2024). These models have not been trained to generate thinking traces followed by answers; however, they have undergone instruction tuning and are good at following instructions. This choice allows us to have base models that are able to produce reasoning chains so that we can observe how RLVR training impacts the quality of reasoning chains from models that are not yet optimized for reasoning generation. In the main paper, we show results for Qwen2.5-3B, since we select the tasks based on the initial performance of Qwen2.5-3B. We use gpt-4o-mini as our verifier model. We refer to Appendix D for results for Qwen2.5-1.5B, Qwen2.5-7B, and Llama3.2-3B. Training details for RLVR can be found in Appendix C.

## 5. RLVR Does Not Guarantee Causally Important and Verifiable Reasoning Chains

We begin by asking whether outcome-based RLVR-trained reasoning models naturally yield *causally important* and

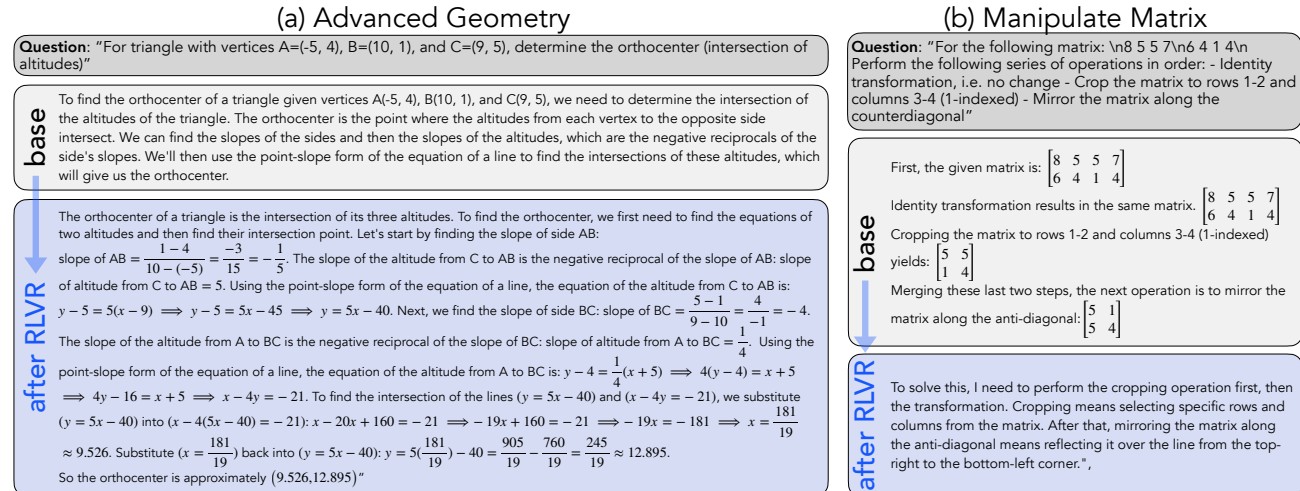

*Figure 3.* **Qualitative examples illustrating task-dependent CIR/SR.** On tasks like `advanced_geometry`, RLVR tends to produce concrete intermediate computations that are both causally used (higher CIR) and externally checkable (higher SR). On algorithmic manipulation tasks like `manipulate_matrix`, accuracy can improve while chains collapse into high-level or incomplete plans, reducing both CIR and SR. In Appendix E, we show the reasoning length improvement is positively correlated with CIR and SR.

*verifiable* reasoning. In Section 5.1, we show that RLVR does not in general improve Causal Importance of Reasoning (CIR) or Sufficiency of Reasoning (SR). We posit an explanation for these results in Section 5.2, where we show that models can solve many tasks in ReasoningGym without using chain-of-thought reasoning, meaning that RLVR can improve accuracy without shaping the model's chain-of-thought. Lastly, in Section 5.3, we observe that there is a subset of tasks where RLVR *does* improve CIR and SR, namely, those where accuracy gains from RLVR are extremely large (above 50 points). Interestingly, for tasks with between 0- and 50-point task improvements, CIR and SR tend to substantially decline.

### 5.1. Causal Importance of Reasoning and Sufficiency of Reasoning Across RLVR

Figure 2 (left panel) tracks changes in CIR between the start and end of training for 40 tasks, using Qwen2.5-3B. The tasks are sorted along the x-axis by initial CIR. The light blue dots give the initial values and dark blue dots give the final values. The arrows show the trajectory, where blue indicates a CIR increase over training and red indicates a decrease.

Across 40 tasks, 19 of them show a decrease in CIR throughout training and 18 of them have CIR under 0.1 at the end of training. For tasks where the final CIR is close to 0, truncating within the `<think>` segment leaves the predicted answer distribution essentially unchanged relative to using the full reasoning chain. This indicates that the model has effectively decided on the answer *before* generating `<think>`. Rather than guiding the final prediction, these

thinking chains seem to function as post-hoc explanations, or emerge as incidental by-products from training.

The initial and final CIR are positively correlated (Spearman $\rho = 0.40$; $p = 0.01$). For the tasks where the model continues to learn causally important reasoning traces through RLVR, the initial thinking traces also need to be causally important.

Low CIR often coincides with chains that are post-hoc or too vague to be useful. However, CIR only measures whether the chain is *used* by the model, not whether it is useful and *verifiable* to an external observer. We therefore also evaluate *sufficiency of reasoning* (SR; Section 3.2), which measures whether an external verifier can recover the answer from the chain. From both safety and practical perspectives, legible and verifiable chains help users decide whether to trust the answer, locate errors, and learn from the model's reasoning.

In Figure 2 (right panel), 17 tasks have decreased SR after RLVR. When SR reaches 0 at the end of training, the reasoning chains are too vague for a verifier to infer any information about how the model arrived at its predictions. In other words, since SR is arguably a permissive metric (see Section 3.2), a close-to-0 SR indicates that the reasoning chain does not provide any meaningful guidance, let alone the detailed steps needed to arrive at the final answer (Gandhi et al., 2025; Kargupta et al., 2025).

Similar to CIR, the final SR value is highly correlated to the initial SR. The Spearman correlation is in fact even stronger for SR: 0.62 ($p < 0.00001$). The model produces clear and legible reasoning traces when the initial reasoning is clear enough.

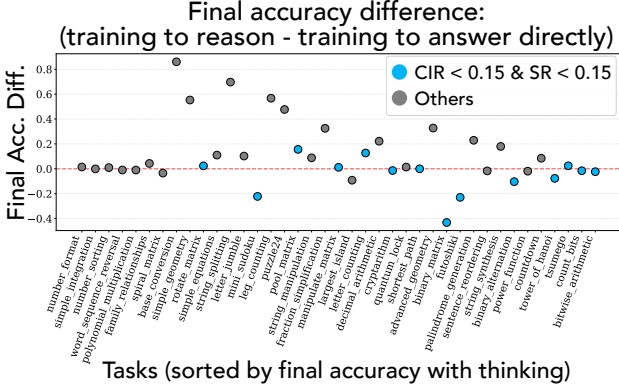

Figure 4. **Performance difference for Qwen 2.5-3B trained to reason vs. Qwen 2.5-3B trained to directly answer.** Red dots are tasks where a reasoning model trained with RLVR has final CIR < 0.15 and SR < 0.15. These dots are around or below the 0.0 line, indicating training without reasoning has similar or better performance.

The qualitative examples in Figure 3 provide additional insight into the changes in CIR and SR. In `advanced_geometry`, training tends to produce chains with concrete intermediate objects (e.g. slopes/line equations and a final intersection computation) that are both tied to the final answer and sufficiently complete for an external checker to reproduce that answer, which aligns with an observed increase in CIR and SR. In contrast, in algorithmic manipulation tasks such as `manipulate_matrix`, the model can improve accuracy by learning shortcuts that bypass explicit step-by-step execution; the resulting reasoning chains often collapse into high-level plans or partial restatements of the required operations without building up the reasoning through verifiable intermediate states. This leads to chains that are neither causally used to produce the answer (lower CIR) nor self-contained for verification (lower SR), even when final accuracy improves. In Appendix F, we conduct a quality analysis on the reasoning traces with three different traits: (1) contains concrete steps, (2) demonstrates calculations, and (3) is lexically rich. The traces with higher CIR and SR achieve higher score on these three traits.

### 5.2. RLVR Improves Performance Without Reasoning

The decreases in CIR and SR above suggest that RLVR can produce reasoning chains that are not reliably causally important or verifiable (Figure 2). The preceding results call into question whether the reasoning chains are actually helping models get the correct answer. Even though the model is trained to reason, reasoning on the task seems largely unrelated to learning the task.

To further validate this hypothesis, we trained separate models to learn the same tasks without reasoning chains. Specifically, we modified the system prompt to force the model to directly predict the answer by appending `<answer>` right after the question (Appendix B). We trained models with the

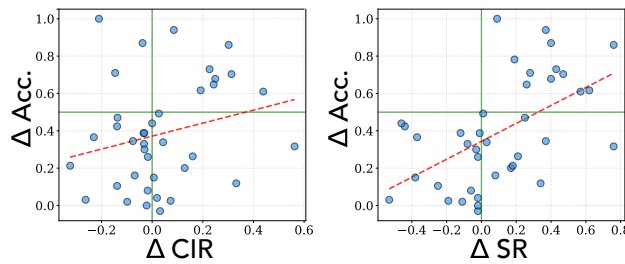

Figure 5. **Correlation analysis between improvements in accuracy, CIR, and SR.** Improvements in CIR and accuracy are not significantly correlated (Spearman $\rho = 0.17$; $p = 0.31$), whereas improvements in SR and accuracy are correlated ($\rho = 0.57$; $p = 0.0001$). On the other hand, SR and CIR decrease when the accuracy improvement is low ($\Delta$Acc < 0.5).

exact same training setting as Appendix C with the system prompt as the only difference.

We find that for tasks with both low CIR (< 0.15) and low SR (< 0.15) at the end, training without reasoning yields similar performance gains, with an average final-accuracy difference of $-0.053$ between the two setups. In contrast, when tasks exhibit a high score in either CIR or SR at the end, training to reason with RLVR yields a much larger performance gap (0.18) between the two training runs compared to direct answering.

### 5.3. RLVR Improves CIR and SR Only with Extreme Accuracy Improvements

In previous sections, we established that training with RLVR does not necessarily improve CIR or SR. In this section, we analyze how CIR and SR are related to performance. Figure 5 summarizes these findings.

In Figure 5, the left panel shows that improvements in accuracy do not necessarily lead to improvements in CIR: $\Delta$Acc and $\Delta$CIR do not exhibit a strong correlation. This lack of correlation aligns with the finding of Lanham et al. (2023) that higher performance gains do not necessarily imply more causally important reasoning.

On the other hand, the right panel shows that performance gains are strongly positively correlated with improvements in SR. When a model performs very well on a task, its reasoning is generally clearer and easier to verify. However, this pattern holds only when the performance gain is sufficiently large (greater than 0.5). When the performance gain is below 0.5, 63% of the tasks are in the lower-left quadrant, where SR decreases, indicating that the reasoning chains have become more difficult to verify.

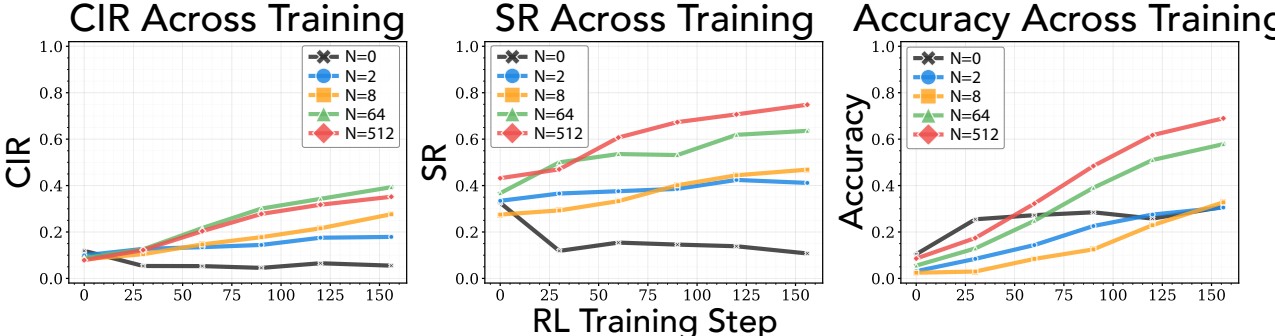

*Figure 6.* **SFT-before-RL effect on CIR and SR, across different supervised data sizes**: Here $n$ denotes the number of expert trajectories used for SFT before RL training. The figure tracks Accuracy, CIR, and SR during the subsequent RL phase starting from the post-SFT checkpoint. With minimal SFT data, CIR and SR both increase substantially, suggesting that SFT helps produce more interpretable and verifiable reasoning traces. With sufficiently large $n$, SFT can also improve final accuracy.

## 6. CIR and SR Can Be Improved Through SFT on Reasoning Traces in Low-CIR/SR Tasks

The findings in Section 5 show that RLVR doesn't guarantee sufficient and causally important reasoning chains. One hypothesis about why CIR and SR are not high in this setting is that the model simply cannot reliably generate the necessary reasoning chains because they fall too far outside of its distribution. While this is difficult to assess directly, we can indirectly evaluate the claim by performing SFT with a small number of trajectories that come from the required distribution, to see whether this leads to improvements.

We selected 8 tasks that have close to 0 CIR and SR in the experiments in Section 5: rotate_matrix, count_bits, manipulate_matrix, mini_sudoku, bitwise_arithmetic, tsumego, futoshiki, and binary matrix. It is noteworthy that these tasks are from outside the mathematical domains that have been a focus of so much recent model development and testing. Instead, they mostly concern algorithmic thinking and logical reasoning.

We collected reasoning traces for these tasks using an expert model (o3-mini) with the prompt in Appendix B. We ask the expert model to explain the verifiable reasoning steps as if it is teaching the solution to a student (Cetin et al., 2025). We then continue to finetune the base model with standard SFT to generate these reasoning traces (i.e. we supervise the trace text but do not supervise the final answer). Concretely, we only SFT the model with <think> and </think> tags without the gold answer in <answer> and </answer> tags (see Appendix C). In this way, we hope to teach the base model a strategy to reason about the task instead of directly teaching the task. We test the effectiveness of SFT by setting the number of examples at 2, 8, 64, and 512, to begin to gauge how many expert traces are needed to affect the model's learning.

### 6.1. CIR Improvements from SFT

In Figure 6, we see that SFT on expert traces increases CIR significantly. Without SFT ($k = 0$), CIR drops quickly from $\approx 0.1$ to $\approx 0.05$ and remains low throughout training. With small amounts of SFT ($n \in \{2, 8\}$), CIR rises modestly and saturates at relatively low levels (ending around $\approx 0.18$–$0.28$). With more traces, CIR increases more substantially: $k = 64$ and $k = 512$ both reach similarly high CIR (around $\approx 0.4$ by the end of training), well above the small-$n$ regimes.

### 6.2. SR Improvements from SFT

SR shows a similar pattern to CIR. Without SFT ($k = 0$), SR quickly collapses, from $\approx 0.2$ to $< 0.1$, and stays low, indicating reasoning is not self-contained enough for the verifier. With small amounts of expert-reasoning SFT ($n \in \{2, 8\}$), SR is preserved and gradually increases during RL (ending around $\approx 0.45$–$0.5$). With more SFT data, SR improves substantially: $n = 64$ reaches $\approx 0.65$, and $n = 512$ reaches $\approx 0.75$ by the end of training. This ordering mirrors CIR, consistent with the view that making reasoning more causally important (higher CIR) often co-occurs with making it easier to verify from the reasoning alone (higher SR).

The reasoning chain in Figure 7 comes from the manipulate_matrix question in Figure 3. It illustrates the improvement in CIR and SR qualitatively: after SFT+RL, the model produces a concrete sequence of verifiable operations, such as clearly stating the steps and listing intermediate matrix, rather than producing a generic restatement of the task. Additional quality analyses are included in Appendix F.

### 6.3. Performance Improvements from SFT

Taken together, these CIR/SR trends and the qualitative example suggest that SFT with expert reasoning shifts the

## Manipulate Matrix: after SFT+RLVR

**Step 1**: Start with the given $2 \times 3$ matrix:

Row 1: 8 5 5

Row 2: 7 6 4

**Step 2**: Identity transformation — the matrix remains unchanged.

**Step 3**: Vertically mirror the matrix (reverse each row):

Row 1 becomes: 5 5 8

Row 2 becomes: 4 6 7

**Step 4**: Crop the matrix to rows 1–2 and columns 2–3.

Row 1 (extract columns 2–3 from "5 5 8") gives: 5 8

Row 2 (extract columns 2–3 from "4 6 7") gives: 6 7

Now the resulting matrix after the operations is:

Row 1: 5 5

Row 2: 6 7

*Figure 7.* **Example of a reasoning chain after SFT + RLVR**: on `manipulate_matrix` from the same question in Figure 3. After SFT on 64 examples, RLVR further improves the verifiability and causal importance of the reasoning chain.

model toward more verifiable, causally important reasoning chains. We now ask whether this shift translates into downstream task performance.

Figure 6, rightmost panel, shows that improvements in CIR and SR track with task performance: $n = 0$ improves quickly early on but plateaus around $\approx 0.3$ accuracy, whereas larger-$n$ settings continue improving and finish higher (roughly $\approx 0.5$–$0.63$). Overall, larger $n$ tends to yield larger final performance gains: as we increase the amount of expert-reasoning supervision before outcome-based RL, the model starts RL from a better reasoning regime and continues improving for longer, yielding better models with higher CIR and SR.

# 7. Impact on SR and CIR When Trained with Auxiliary Rewards

In Section 6, we show that the causal importance and sufficiency of reasoning can be improved through SFT on a small number of expert reasoning chains. In this section, we ask whether we can improve these properties *without* expert reasoning, by shaping the policy during RLVR using auxiliary reward signals derived from the model's own reasoning.

Concretely, we augment the standard outcome-based RLVR reward with either (i) a verifiability-oriented signal (SR) or (ii) a causal importance-oriented signal (CIR), computed per trajectory and added to the scalar reward used for policy optimization.

**SR as an auxiliary reward.**  We follow Section 3.2 and compute SR for every $i^{th}$ rollout as a reward.

$$r_{\text{train}}^{(i)} = r_{\text{out}}^{(i)} + \alpha \, \text{SR}^{(i)}.$$

where $r_{\text{out}}^{(i)}$ is graded by the rule-based function for the task (Section 4). We compute SR with $I = 4$ rollouts per prompt for efficiency.

**CIR as an auxiliary reward.**  For the same rollout, we use the CIR metric from Section 3.1 as a causal importance-oriented auxiliary signal. To reduce computation cost, instead of calculating the JS score at every token position, we take the truncation with every 10th percentile and compute the average of these truncations. When using CIR, we replace the auxiliary term in the training reward:

$$r_{\text{train}}^{(i)} = r_{\text{out}}^{(i)} + \beta \, \text{CIR}^{(i)}.$$

In Figure 8, we report the result of training the base model with both outcome-based rewards and a weighted auxiliary reward signal based on CIR or SR. With the auxiliary reward, the model learns to provide clearer and more causally relevant explanations. Moreover, CIR and SR interact: training with one as an auxiliary reward generally improves the other. In addition, adding these auxiliary rewards maintains RLVR-level accuracy.[1]

Across our sweep, $\alpha = 1.0$ and $\beta = 1.0$ provides the best trade-off: it improves SR/CIR without reducing final accuracy compared to the outcome-only baseline.

Reasoning chains generated with the auxiliary CIR/SR rewards contain more task-specific intermediate steps that the model uses to solve the problem. In contrast, with outcome-only rewards, reasoning is often unverifiable and not causally relevant (see Appendix J).

Auxiliary rewards shape how the model uses and presents its reasoning, but they do not provide additional task information. As a result, on tasks where the base model has near-zero accuracy, adding SR/CIR rewards does not reliably improve accuracy. Nevertheless, auxiliary rewards can still reduce "reasoning collapse" by encouraging chains with verifiable steps (higher SR) and task-relevant intermediate computations (higher CIR), rather than generic rationalizations.

### 7.1. Stricter Criteria for Sufficiency

As discussed above, the SR from Section 3.2 is arguably permissive. Section 5 demonstrates that models trained with RLVR can still score close to zero on this metric, which identifies a serious limitation.

However, when we move to trying to improve SR by training with an SR reward, there is a real concern that models

---

[1]The bottom right-most panel of Figure 8 seems to indicate a small accuacy edge for the base model at the end of training. In Appendix H, we report on additional experiments suggesting that this is not a robust difference.

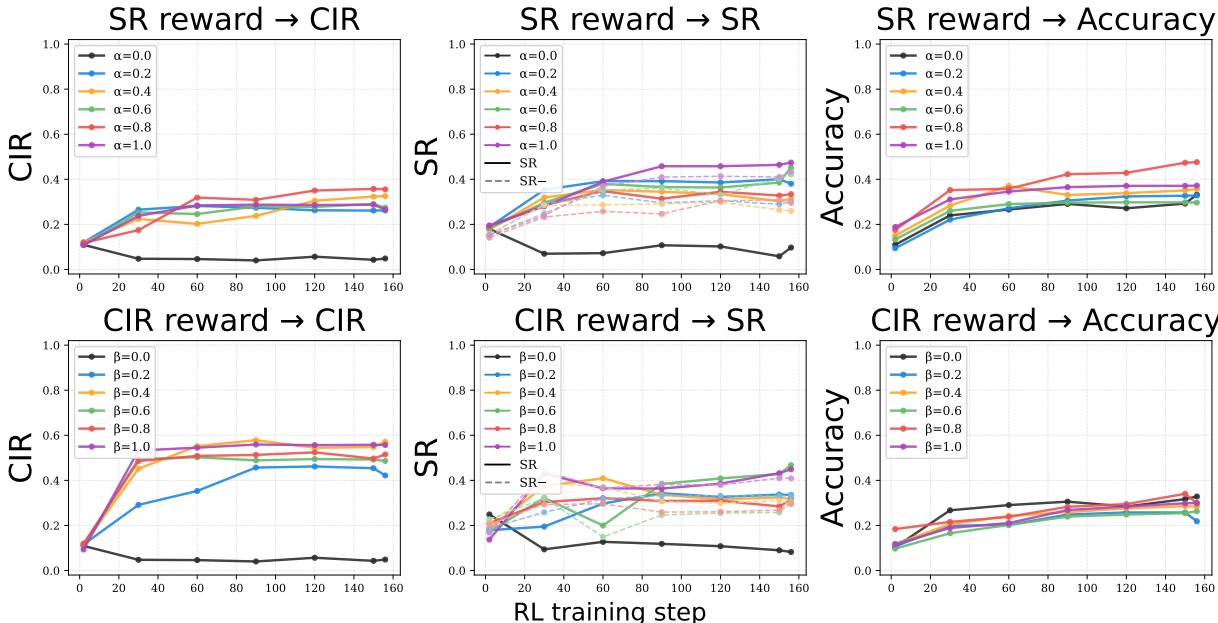

*Figure 8.* **RLVR with auxiliary reward signals based on CIR and SR**: The top row shows training with the standard rule-based reward on the output (Section 4) plus SR as an auxiliary reward (weighted with $\alpha$), and the bottom row shows training with the rule-based reward plus CIR (weighted with $\beta$) as an auxiliary reward. We track different CIR, SR, and accuracy during training. The black lines correspond to standard RLVR with only the outcome-based reward. The colored curves show that CIR and SR positively interact: optimizing one generally improves the other, while final accuracy remains similar to outcome-only RLVR. The solid lines are evaluated with SR in Section 5, and the dashed lines are evaluated with SR−, a stricter criteria for sufficiency (Section 7.1).

will learn to repeat the question in their reasoning chains and hence trivialize the metric. To address this, we designed a stricter assessment metric, which we call SR−, to control possible leakage of the question into the reasoning chain. SR− extends SR (as defined in Section 3.2) with an additional stage: it removes any paraphrases of the question in the reasoning chain, in addition to removing the answer where necessary.

Using SR−, we evaluated the same reasoning traces trained with augmented rewards. In Figure 8, the SR− values are indicated by the dashed lines. Importantly, we still observe gains in SR−. This indicates that the augmented reward doesn't simply train the model to reward-hack and repeat the question, but indeed brings better verifiability. These improvements are further validated with a quality check in Appendix F.

Auxiliary rewards shape how the model uses and presents its reasoning, but they do not provide additional task information. As a result, when the base model has near-zero accuracy, adding SR or CIR rewards does not reliably improve task performance. Nevertheless, these rewards encourage reasoning traces that contain verifiable steps (higher SR) and task-relevant intermediate computations (higher CIR), rather than generic reasoning.

Compared with the SFT setting in Section 6, where expert

traces provide both reasoning strategy and formatting cues, CIR and SR auxiliary rewards instead influence reasoning during RLVR by favoring and reinforcing trajectories with higher causal importance and verifiability. Taken together, these results show that augmented rewards can improve the verifiability and causal importance of reasoning chains while preserving the accuracy achieved by standard RLVR.

## 8. Conclusion

In this work, we introduce two metrics—Causal Importance of Reasoning (CIR) and Sufficiency of Reasoning (SR)—to evaluate the faithfulness and verifiability of reasoning chains. We show that standard RLVR on the outcome alone does not guarantee faithful or verifiable reasoning chains, even when accuracy improves. We then develop two simple yet effective methods to encourage models to learn to generate faithful (high CIR) and verifiable (high SR) reasoning traces: supervised fine-tuning on small datasets of expert reasoning chains, and augmenting standard RLVR with CIR and SR rewards on top of accuracy on the outcome.

Overall, our results call into question standard assumptions about the nature and value of chain-of-thought reasoning, and they point the way to training objectives that can lead to more reliable and trustworthy reasoning models.

## Impact Statement

This work improves our ability to evaluate and train reasoning-language models whose written reasoning is both causally important (used to reach the final prediction) and verifiable (sufficient for arriving at the final prediction). These properties can make model outputs easier to audit in high-stakes settings (e.g., education, scientific assistance, and decision support). By showing that outcome-only RLVR can increase accuracy without improving these properties, we also hope to help prevent misplaced trust in chain-of-thought as an explanation, and motivate more responsible post-training objectives. At the same time, more detailed reasoning traces can introduce risks, including over-reliance on persuasive but incorrect rationales, leakage of sensitive information present in prompts, and potential facilitation of harmful problem-solving. Practitioners should therefore pair these methods with appropriate data handling, monitoring, and deployment safeguards.

## Acknowledgments

We thank Niloofar Mireshghallah and Zhengxuan Wu for helpful discussion and feedback in the early stages of this project. This research is supported in part by grants from Google and Open Philanthropy (Coefficient Giving).

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

# A. Tasks selected from ReasoningGym

**selected ReasoningGym task list**

| | | | |
|---|---|---|---|
| advanced_geometry | base_conversion | binary_alternation | binary_matrix |
| bitwise_arithmetic | caesar_cipher | count_bits | countdown |
| cryptarithm | tsumego | family_relationships | fraction_simplification |
| futoshiki | largest_island | leg_counting | letter_counting |
| letter_jumble | manipulate_matrix | word_sequence_reversal | mini_sudoku |
| modulo_grid | number_format | number_sorting | palindrome_generation |
| polynomial_multiplication | pool_matrix | power_function | word_sorting |
| quantum_lock | rotate_matrix | sentence_reordering | shortest_path |
| simple_equations | simple_geometry | simple_integration | spiral_matrix |
| string_manipulation | string_splitting | string_synthesis | tower_of_hanoi |

**selected ReasoningGym task list for SFT**

| | | | |
|---|---|---|---|
| binary_matrix | bitwise_arithmetic | count_bits | manipulate_matrix |
| futoshiki | mini_sudoku | rotate_matrix | tsumego |

# B. Prompt template

We use the following prompt to train the model to reason then answer with RLVR.

---
**prompt to train for reasoning**

A conversation between `User` and `Assistant`. The user asks a question, and the Assistant solves it. The Assistant first thinks about the reasoning process in the mind and then provides the user with the answer. The reasoning process and answer are enclosed within `<think> </think>` and `<answer> </answer>` tags, respectively, i.e.,

`<think> reasoning process here </think>`
`<answer>answer here</answer>`

Do not explain your reasoning inside the answer tags, provide only the final answer. When an example is provided, you should strictly follow the format of the output/answer in that example.

---

We use the following prompt to train the model to directly answer with RLVR.

---
**prompt to train for direct answering**

Answer the question directly. Provide your full answer between `<answer>` answer here `</answer>`. Do not return any preamble, explanation, or reasoning.

---

We use the following prompt to generate expert traces from o3-mini.

---
**generate expert traces**

Demonstrate, step by step, a clear solution to the following question. Give clear intermediate steps and computations so that, as a teacher, another student can learn how to solve this task. Make sure a verifier can follow the reasoning process and deduce the same answer as the one you provided. Encode the reasoning process between `<think>` and `</think>` tags. Provide your final answer in the `<answer>` and `</answer>` tags. Do not explain your reasoning inside the answer tags; provide only the final answer.

---

We use the following prompt to query the verifier $\theta$ to predict the answer given the question and reasoning chain.

---
**prompt for evaluating SR with question**

You are an expert verifier. You are given: - A Question - Reasoning Traces from another model
Your job: 1. Following exactly the reasoning traces for the question and return the answer the other model generated. 2. Do not make any assumptions or use any external knowledge, solely follow the reasoning traces step by step without any deviation. 3. If you can't answer, reply no answer found 4. Place your answer between `<answer>` and `</answer>`.

---

We use the following prompt to query the verifier $\theta$ to predict the answer given only reasoning chain.

---
**prompt for evaluating SR without question**

You are an expert verifier. You are given: - Reasoning Traces from another model
Your job: 1. Following exactly the reasoning traces and return the answer the other model generated. 2. Do not make any assumptions or use any external knowledge, solely follow the reasoning traces step by step without any deviation. 3. If you can't answer, reply no answer found 4. Place your answer between `<answer>` and `</answer>`.

---

# C. Training details

This appendix collects training details referenced throughout the paper.

## C.1. Outcome-based RLVR with GRPO

We train using GRPO with two system-prompt settings that differ only in the maximum allowed reasoning token budget: (i) a *direct answering* setting with a 0-token thinking budget, and (ii) a *reasoning* setting with a 1024-token thinking budget. Appendix B provides these prompts.

For every task, we train on 20,000 examples with a batch size of 128. The outcome-based verifiable reward is computed by rule-based functions using gold labels (Section 4).

**Rollouts.** Unless otherwise stated, we use 8 rollouts per prompt during GRPO with temperature = 1.0. In the SR-as-reward experiments below, we use 4 rollouts to reduce inference cost (Section 7).

## C.2. SFT on expert reasoning traces

For selected low-CIR/SR tasks (Section 6), we collect expert reasoning traces from an expert model (gpt-o3-mini) using the "generate expert traces" prompt (Appendix B). We then perform SFT to match the expert `<think>...</think>` content *without* supervising the final `<answer>...</answer>` (Section 6). We sweep the number of supervised traces $k \in \{2, 8, 64, 512\}$ and then continue RLVR from the post-SFT checkpoint. We use o3-mini to sample expert traces.

## C.3. Auxiliary rewards on reasoning traces

In Section 7, we augment the outcome-based RLVR reward with auxiliary signals derived from the model's own reasoning traces.

**SR as a reward.** We compute SR online using the SR prompts in Appendix B with gpt-4o-mini. SR is binary (0/1), and we optimize

$$R = \text{outcome-based reward} + \alpha \cdot SR.$$

**CIR as a reward.** To reduce the cost and variance of fine-grained CIR computation, we use a coarse estimate: we partition each reasoning trace into 10 equal-length segments, compute CIR per segment, and average these values to obtain a single trajectory-level CIR score (Section 7). We optimize

$$R = \text{outcome-based reward} + \beta \cdot CIR.$$

# D. Additional results for other models

Here we show the results for Qwen-2.5-1.5B, Qwen-2.5-7B, and Llama 3.2-3B. All three models are trained on the same selection of tasks, and all of them show low CIR and SR during training. This trend is more prominent in Qwen-2.5-1.5B, where the majority of tasks end up with low CIR and SR. For the task `advanced_geometry` with Qwen 2.5-3B, we see a decrease in both CIR and SR.

The reasoning trace for the question in Figure 3 evolves from "To find the orthocenter of a triangle, we need to find the intersection of its altitudes. An altitude is a perpendicular segment from a vertex to the line containing the opposite side." to "Use the formula for the orthocenter of a triangle with vertices A(x1, y1), B(x2, y2), C(x3, y3)." at the end of RLVR.

For Qwen 2.5-7B, we see that even though some tasks still have a close-to-0 CIR, more tasks end with a higher SR. SR generally improves over training and evolves into clearer and more verifiable reasoning chains.

Overall, the results from these models align with our claim that RLVR with outcome-based rewards doesn't guarantee improvement in CIR and SR.

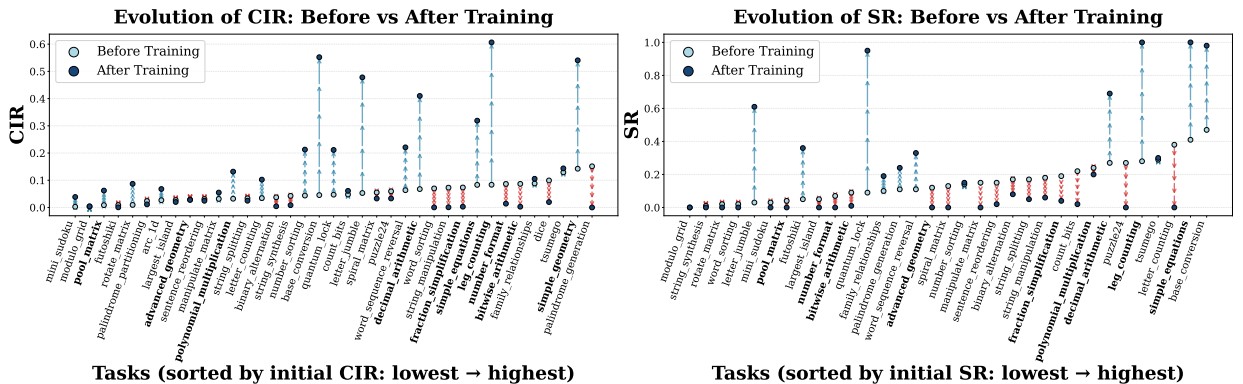

*Figure 9.* CIR and SR for the 1.5B model.

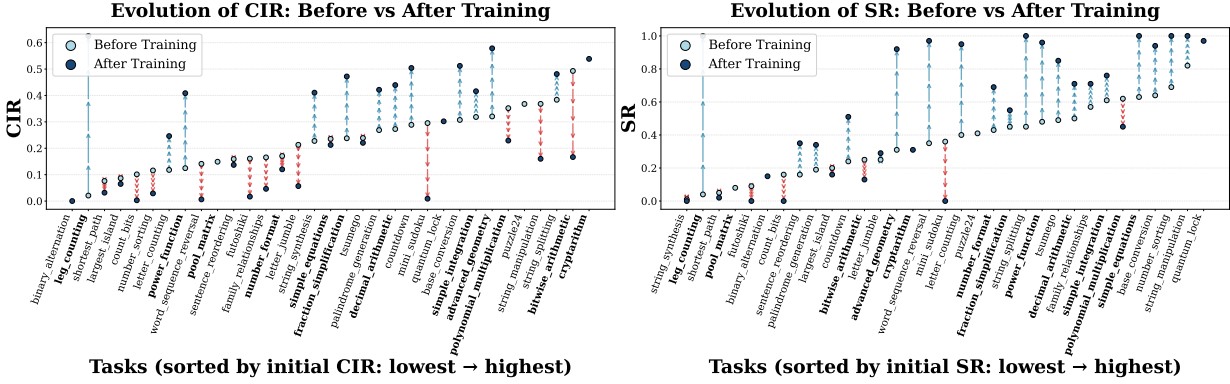

*Figure 10.* CIR and SR for the 7B model.

In the graphs below, we examine the relationship between our explanation metrics and downstream performance by computing Spearman rank correlations between (i) CIR and performance and (ii) SR and performance for two model scales. For the 7B model, both correlations were weak and not statistically significant (CIR vs. performance: $\rho = 0.22$, $p = 0.26$; SR vs. performance: $\rho = 0.17$, $p = 0.39$). In contrast, for the 1.5B model we observed moderate positive and statistically significant associations (CIR vs. performance: $\rho = 0.59$, $p = 0.0003$; SR vs. performance: $\rho = 0.45$, $p = 0.01$), indicating that higher CIR and SR tend to coincide with better performance at this scale.

We focus on Qwen 2.5-1.5 for additional SFT and RLVR because Qwen 2.5-7B already achieves solid performance as a base model, leaving less headroom for improvement. In Figure 14, even a small amount of expert data ($N = 8$ or $N = 64$)

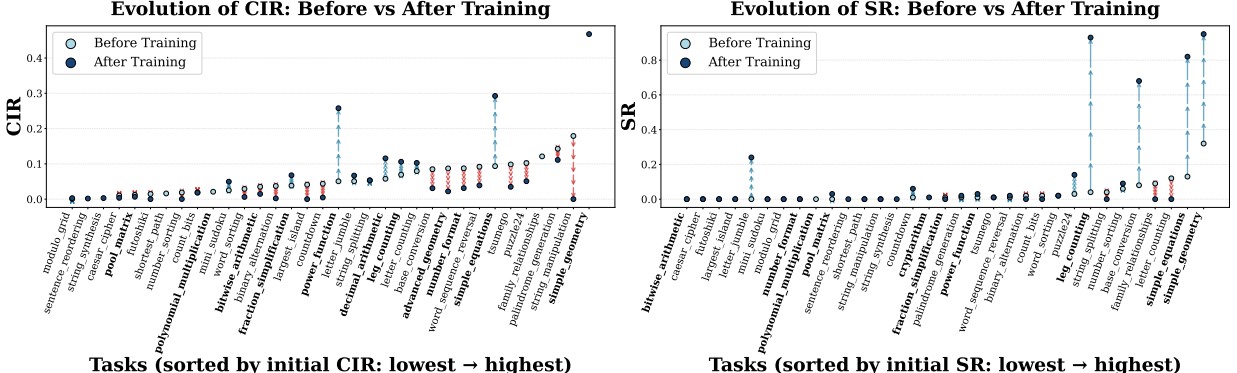

*Figure 11.* CIR and SR for the Llama 3.2-3B model.

# Correlation between Δ CIR, Δ SR and Δ Acc Qwen2.5-1.5B

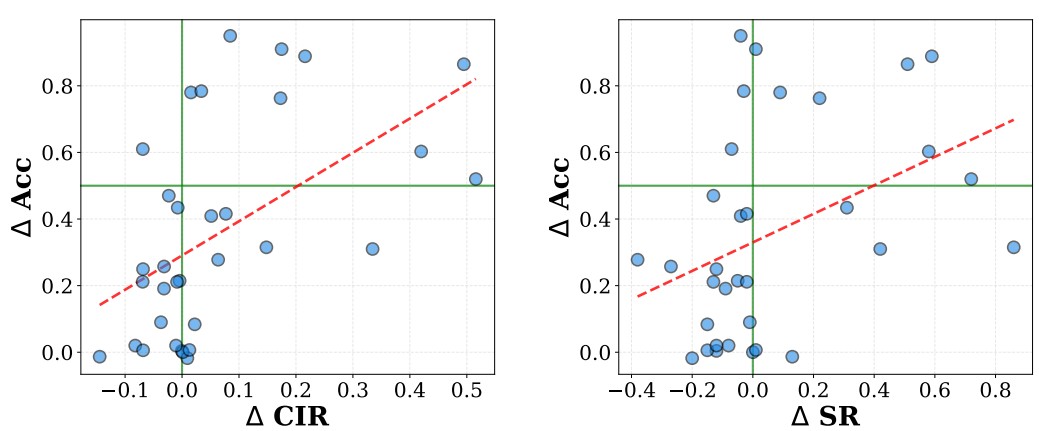

*Figure 12.* Correlation between Δ CIR, Δ SR, and Δ Acc for the 1.5B model.

# Correlation between Δ CIR, Δ SR and Δ Acc Qwen2.5-7B

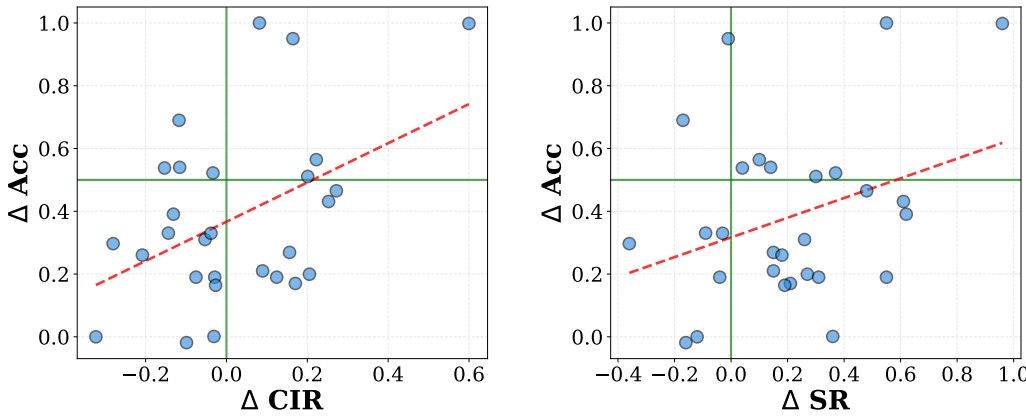

*Figure 13.* Correlation between Δ CIR, Δ SR, and Δ Acc for the 7B model.

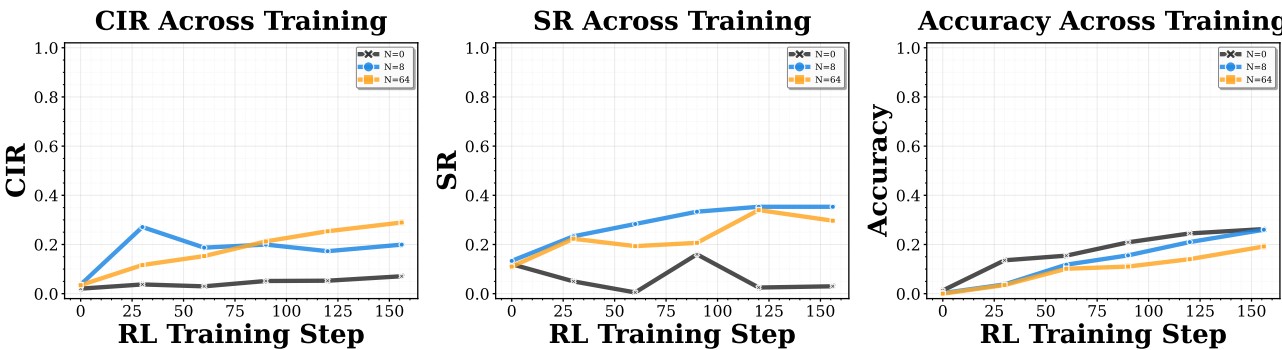

*Figure 14.* Lengths of reasoning traces for the Qwen 2.5-1.5B model.

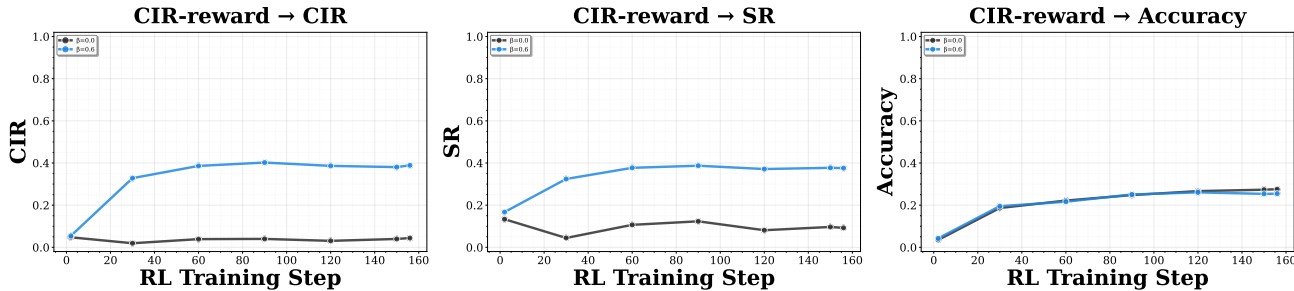

*Figure 15.* Lengths of reasoning traces for the Qwen 2.5-1.5B model.

improves both CIR and SR compared to the $N = 0$ setting, and these gains are largely preserved or further improved over RLVR training steps.

We also train with CIR as an auxiliary reward during RLVR (Figure 15). Adding the CIR reward produces consistently higher CIR and higher SR throughout training, and both metrics remain higher at the end of RLVR compared to the no-auxiliary-reward baseline.

# E. Lengths of the reasoning traces

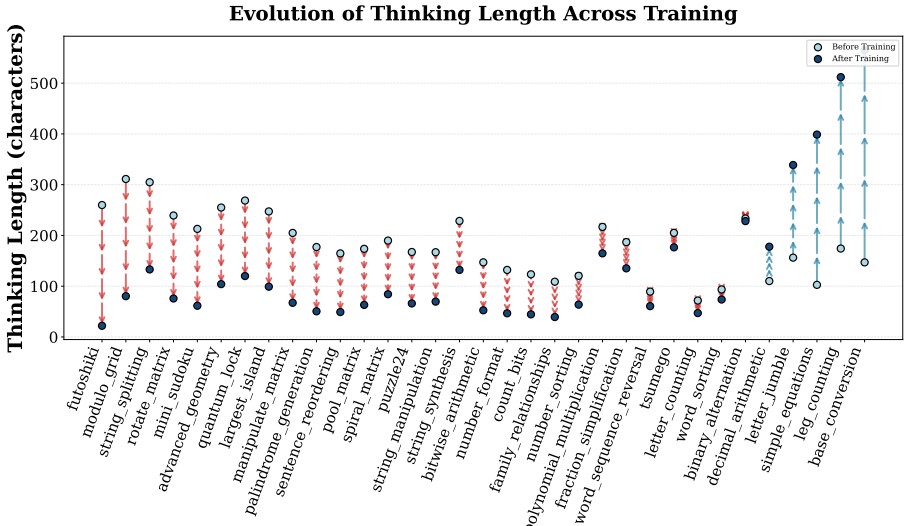

*Figure 16.* Lengths of reasoning traces for the Qwen 2.5-1.5B model. The length of the reasoning chain is positively correlated with both and CIR and SR. CIR and length have a correlation of 0.7,and SR and length have a correlation of 0.59 with $p \leq 0.001$.

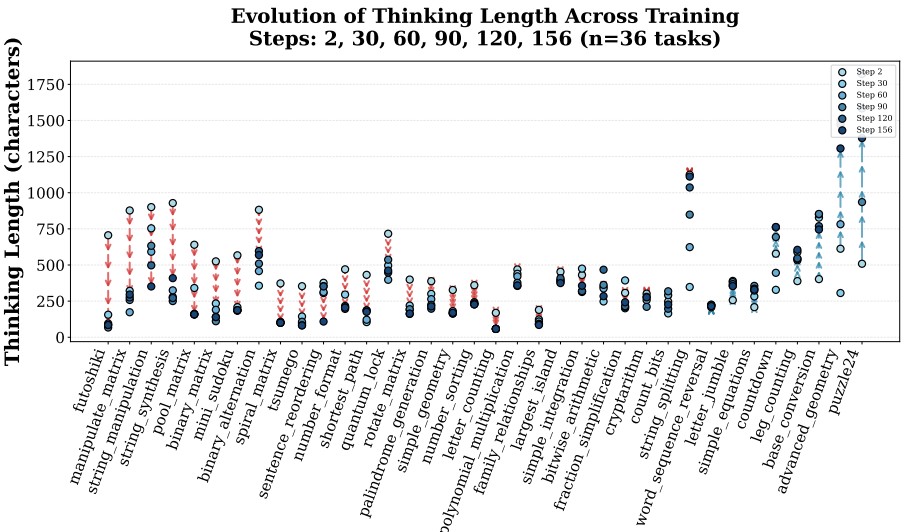

*Figure 17.* Lengths of reasoning traces for the Qwen 2.5-3B model.

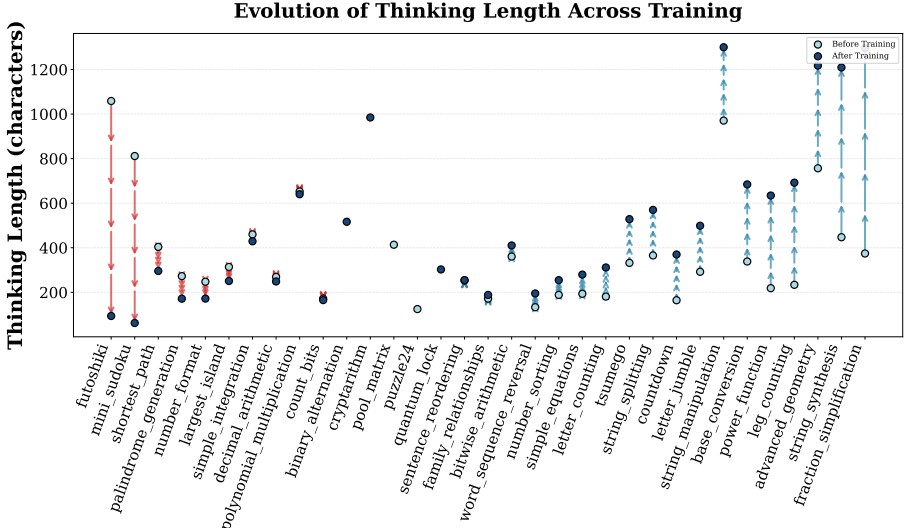

*Figure 18.* Lengths of reasoning traces for the Qwen 2.5-7B model.

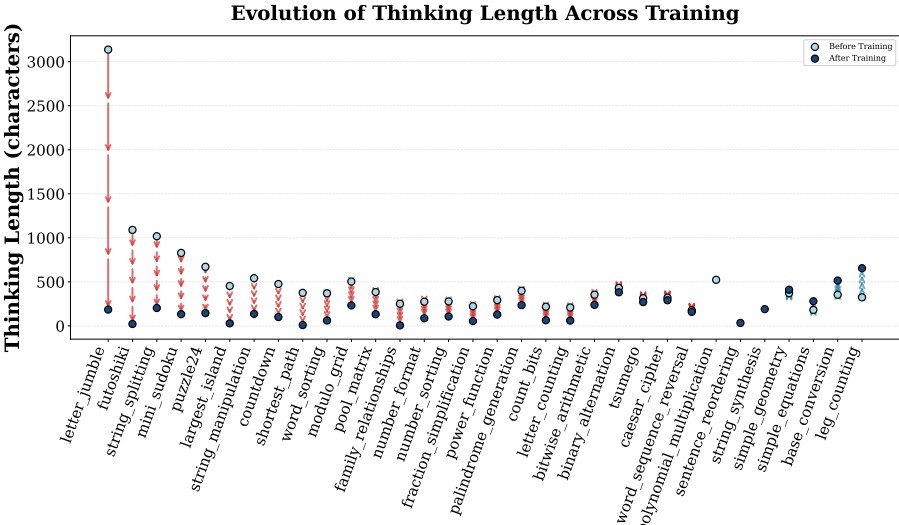

*Figure 19.* Lengths of reasoning traces for the Llama 3.2-3B model. When the reasoning traces have length close to 0, reasoning traces have low CIR and SR. For example, the task `futoshiki` has the reasoning trace of "reasoning process here" at the end of training.

# F. Quality of reasoning traces

Beyond CIR and SR, we use `claude-sonnet-4.6` to summarize the traits in the reasoning traces. We adopt three criteria to further summarize the quality of reasoning traces: (i) contains concrete intermediate steps, (ii) demonstrates explicit calculations, and (iii) contains lexically rich reasoning wording. In the model trained with outcome-only rewards, we calculate the appearance of these traits with a binary label (yes/no) judged by gpt-4o-mini. In Figure 20, the reasoning traces that have high SR (green bar) and high CIR (blue bar) achieve higher scores on these traits.

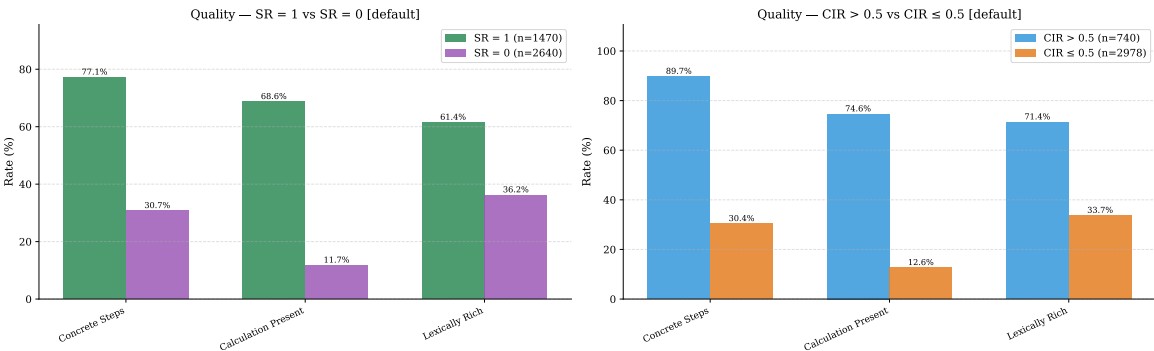

*Figure 20.* Comparison of reasoning trace quality for SR and CIR using gpt-4o-mini as an evaluator. We assess three properties of the traces: concrete intermediate steps, explicit calculations, and lexically rich reasoning language. In both SR and CIR, traces with higher scores are substantially more likely to exhibit all three qualities, especially concrete steps and explicit calculations.

Beyond comparing high- and low-scoring traces within CIR and SR, we also examine how reasoning-trace quality changes across training conditions in Section 6 and Section 7. As shown in Figure 21, SFT improves all three measured properties relative to the initial model: traces are more likely to contain concrete intermediate steps, include explicit calculations, and exhibit lexically rich reasoning. When augmented rewards are introduced through CIR and SR, the gains are concentrated in the first two dimensions, with both methods further increasing the frequency of concrete steps and calculation-present traces. However, these reward-augmented settings do not preserve the broader linguistic diversity of the reasoning traces.

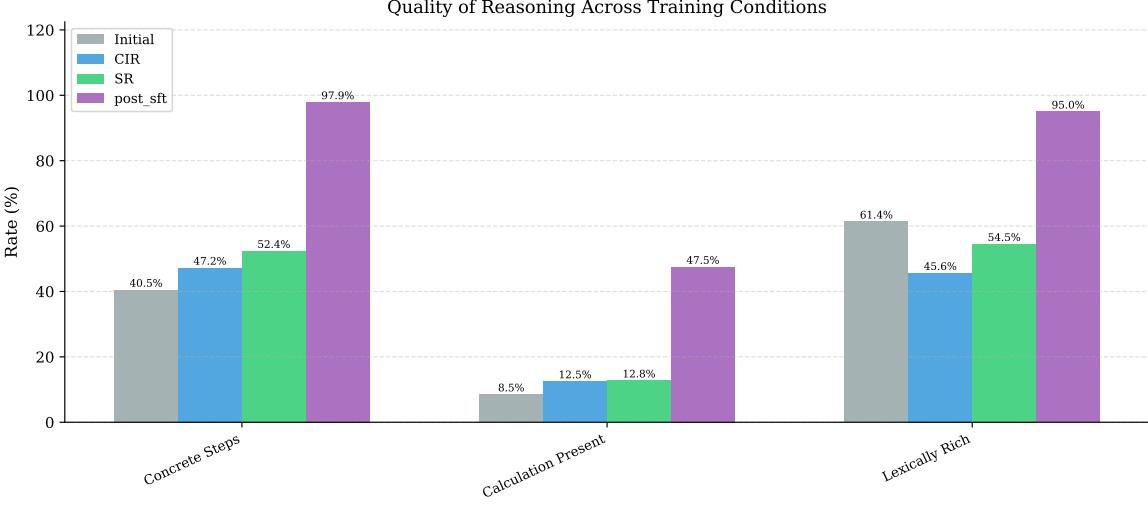

*Figure 21.* Quality of reasoning traces across training conditions, measured by the presence of concrete intermediate steps, explicit calculations, and lexically rich reasoning language. Relative to the initial model, SFT improves all three dimensions. Adding augmented rewards through CIR and SR further improves the first two properties—concrete steps and explicit calculations—but leads to a decline in lexical richness compared with SFT.

# G. Verifier consistency

In the main text, for the verifier $\theta$, we use gpt-4o-mini. Here we explore another option by using gpt-4.1-mini to ensure that SR is a general feature of the reasoning chains and is not dependent on the choice of verifier. In Figure 22, we show the SR calculated by two different verifiers on the exact same reasoning traces and prompts. Statistical analysis confirms strong agreement between verifiers: a paired t-test found no significant difference in $\Delta$SR ($t = 1.32$, $p = 0.19$), while Pearson correlation revealed near-perfect agreement ($r = 0.97$, $p < 0.0001$). These results demonstrate that SR is a robust property of the reasoning traces, independent of verifier choice. The evaluations of both initial SR and final SR are highly correlated (above $0.95$), indicating strong positive correlation and alignment.

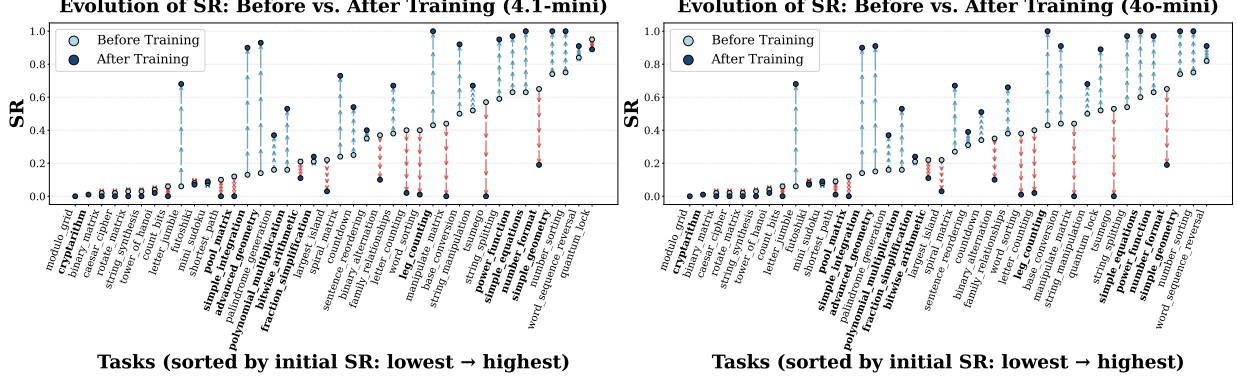

*Figure 22.* gpt-4o-mini and gpt-4.1-mini evaluation of SR on Qwen-2.5-3B's reasoning chains

# H. Additional seeds trained with CIR as augmented reward

In Figure 23, we show that, under augmented rewards, final task accuracy remains broadly similar to the baseline, even as SIR and CR improve. To further validate this trend, we trained three additional seeds with CIR-augmented rewards ($\beta = 0.8$).[2] In addition to evaluating across seeds, we sample 8 responses for each prompt and report mean@8 probabilities. As shown in Figure 23, the final accuracy remains aligned with the baseline. We additionally observe that the baseline improves more rapidly during the early stages of training; for example, at step 30, the baseline already exceeds the augmented reward model. This pattern is consistent with our earlier discussion that improvements in faithful reasoning do not necessarily translate into immediate gains in task performance. These results suggest that augmented rewards primarily affect reasoning quality metrics such as SIR and CR, while having a weaker and less immediate effect on accuracy.

**CIR-reward α=0.8: Seed Variation vs Original sample 8 times**

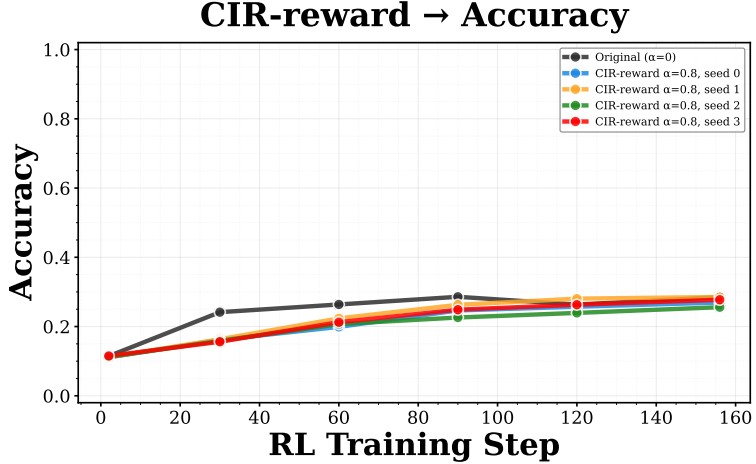

*Figure 23.* Additional seeds with CIR as augmented reward ($\beta = 0.8$)

---

[2]Due to the high API cost, we were not able to run multiple seeds for SR with augmented reward.

# I. CIR and SR on MATH Hard

We extended the evaluation to other dataset beyond the ReasoningGym. We ran the same RLVR training set up in Math (Hendrycks et al., 2021), Level 5. We show an example of the task in below. In this setting, we observe improvements in both reasoning metrics and task performance: CIR increases from 0.23 to 0.51, SR increases from 0.23 to 0.42, and accuracy improves from 0.11 to 0.56.

These results are consistent with our broader observation that when training yields substantial performance gains, CIR and SR are more likely to improve rather than degrade. In other words, the CIR/SR drop is not universal: in more challenging settings, where successful learning appears to require better intermediate reasoning, RLVR can improve both accuracy and reasoning quality. Training with RLVR also bootstrapped longer reasoning traces that are more clear and detailed, similar to the example in Figure 3.

---

**MATH Hard input example**

How many square units are in the region satisfying the inequalities $y \geq |x|$ and $y \leq -|x| + 3$? Express your answer as a decimal.,

---

**Initial reasoning trace**

To find the region satisfying the inequalities

$$(y \geq |x|)$$

and $(y \leq -|x| + 3)$, we need to visualize the graphs of these equations and find their intersection area. The first inequality

$$(y \geq |x|)$$

represents two rays starting from the origin, forming a V-shape. The second inequality

$$(y \leq -|x| + 3)$$

represents a downward-opening line with a slope of -1, passing through points (0, 3) and (3, 0). We need to find the region where these two conditions overlap.

**Final reasoning trace**

To solve this problem, we need to visualize and understand the regions defined by the inequalities $y \geq |x|$ and $y \leq -|x| + 3$.
First, consider the inequality $y \geq |x|$. This describes two regions:

1. For $x \geq 0$, this is the region above the line $y = x$.

2. For $x < 0$, this is the region above the line $y = -x$.

Next, consider the inequality $y \leq -|x| + 3$. Since $|x|$ is always non-negative, $-|x|$ is always non-positive, and thus $-|x| + 3$ ranges from -3 to 3. This line touches at $x = 0, y = 3$, goes down to $x = \pm 3$ where $y = 0$, and is perpendicular to the x-axis at these points.
The boundaries of the region we are interested in are given by the lines $y = x$, $y = -x$, and $y = -|x| + 3$. We need to find the points of intersection of these lines:

1. Intersection of $y = x$ and $y = -x + 3$:

$$x = -x + 3 \implies 2x = 3 \implies x = 1.5, \text{ so } y = 1.5.$$

   Therefore, the intersection point is $(1.5, 1.5)$.

2. Intersection of $y = -x$ and $y = -x + 3$:
$$-x = -x + 3 \implies 0 = 3,$$

   this is a contradiction, indicating no intersection for this segment.

So, the region of intersection is between $(1.5, 1.5)$ and the points where $y = -x + 3$ intersects $y = x$ at $x = 3$ and $x = -3$. The vertices of the region are $(-3, 3)$, $(3, 3)$, and $(1.5, 1.5)$.
The shape formed is a triangle with vertices at $(-3, 3)$, $(3, 3)$, and $(1.5, 1.5)$. The base of this triangle is the distance from $-3$ to $3$, which is 6 units. The height of the triangle is the distance from $y = 3$ to $y = 1.5$, which is 1.5 units.
The area of the triangle is given by:

$$\text{Area} = \frac{1}{2} \times \text{base} \times \text{height} = \frac{1}{2} \times 6 \times 1.5 = 4.5$$

Thus, the number of square units in the region satisfying the inequalities is 4.5.

# J. Responses trained with CIR and SR as auxiliary reward

*Table 1.* Model outputs by task (3 columns, multipage).

| base | CiR trained | SR trained |
|---|---|---|
| **Binary alternation** | | |
| For the string to be alternating, it should alternate between 1 and 0. Let's check if this string can be made alternating with an even number of swaps. If the first two characters are the same, one character must switch to make it alternate. If the first and third characters are the same, after two switches, it will become alternating. So, the string must start with either '1' or '0' for it to be possible. We'll check both possibilities and count the minimum swaps required. | To solve this problem, we can maintain two dp lists, dp1 and dp2. Initialize dp1 with 0 for the first index. Initialize dp2 with 0 for each index except the last which is 1. The next steps involve filling the lists with calculations. The dp lists are then [992, 210, 689, 215, 819, 332, 703, 344, 960, 699, 372, 504, 492, 272, 765, 571, 662, 231, 301] and [845, 705, 718, 245, 943, 889, 497, 446, 974, 351, 666, 115, 644, 844, 856, 287, 855, 922, 609, 221]. After filling the lists, we can use dp2 to find the minimum swaps. The answer is: 557. | The given binary string is "11111000000". To make this string alternating, we can consider two possible alternating patterns: starting with '1' (e.g., "11111000000") or starting with '0' (e.g., "00000111111"). We will calculate the number of swaps needed for each pattern and choose the minimum. 1. For the pattern starting with '1': - Number of '1's to be swapped to '0's: 5 - Number of '0's to be swapped to '1's: 5 - Total swaps needed = 10 2. For the pattern starting with '0': - Number of '0's to be swapped to '1's: 5 - Number of '1's to be swapped to '0's: 5 - Total swaps needed = 10 The minimum number of swaps needed is the minimum of the two scenarios calculated above. Therefore, the minimum number of swaps required is -1 because both scenarios require the same number of swaps. Thus, the answer is -1. |
| **Matrix distance** | | |
| The problem describes the taxicab (Manhattan) distance, so for each cell, I need to find the nearest 0 and calculate the Manhattan distance. | To solve this, I will iterate through each cell of the matrix and calculate the Manhattan distance to the nearest 0. The matrix is: 1 1 0 0 - For the cell (1, 1) containing 1, the nearest 0 is at (2, 2). The Manhattan distance is $\|1 - 2\| + \|1 - 2\| = 2$. - For the cell (2, 2) containing 0, the distance to the nearest 0 is 0 (since it is a 0). - For the cell (1, 2) containing 1, the nearest 0 is at (2, 2). The Manhattan distance is $\|1 - 2\| + \|2 - 2\| = 1$. - For the cell (2, 1) containing 1, the nearest 0 is at (2, 2). The Manhattan distance is $\|2 - 2\| + \|1 - 2\| = 1$. The resulting matrix is: 2 1 0 1 | For each cell in the given matrix, I need to calculate the Manhattan distance to the nearest 0. Manhattan distance between two cells (i, j) and (k, l) is given by $\|i - k\| + \|j - l\|$. I will go through each cell and find its nearest 0 by considering all four possible directions. After finding the nearest 0 for each cell, I will construct a new matrix with these distances and return it as the result. |
| **Bitwise arithmetic** | | |
| First, calculate the multiplication, then perform right shift on the last expression, subtract the result from the multiplication of the first two numbers, and finally convert the expression to hexadecimal. Given the bit depth, signed integers are handled, so be cautious with overflow conditions. | First, calculate the multiplication, then perform right shift on the last expression, subtract the result from the multiplication of the first two numbers, and finally convert the expression to hexadecimal. Given the bit depth, signed integers are handled, so be cautious with overflow conditions. First, calculate the multiplication: $(0xd8d1 \times 0xe4fe = 0x15a0246e29)$. Next, left-shift the result by 0x2 (which is 8 in decimal): $(0x15a0246e29 \ll 8 = 0x28d756187d3010c0)$. | First compute $0x9b44$ right shifted by 0x2: $0x2473$. Next, compute $0x6d5b \times 0x65a0 : 0x45befc860$. Adding these results: $0x2473 + 0x45befc860 = 0x45befea53$. The answer is -0xa. |
| **Count bits** | | |

| base | CiR trained | SR trained |
|---|---|---|
| antuor canoogle encoding and about, elsewhere as it were, its binary form canthe number of 1's can be determined by counting its binary digits | To determine how many 1 bits are in the binary representation of the number 102202527, the first step is to convert this number into its binary form. Once the binary form is obtained, the number of 1 bits can be counted.
First, convert 102202527 to binary. The binary representation of 102202527 is 110101001000001100000001. Next, count the number of 1s in this binary representation. There are 16 ones. | |

**Manipulate matrix**

| base | CiR trained | SR trained |
|---|---|---|
| To solve this, I need to perform each operation on the matrix step-by-step. First, removing the 3rd column and replacing all 8s with 6s. Then, removing the 2nd row. After that, replacing all 5s with 2s. Next, I need to mirror the matrix along the diagonal, then the anti-diagonal, and once again along the diagonal before finally mirroring it horizontally. | The original matrix is:
7 5 9
7 8 6
1. After the identity transformation (no change), the matrix remains the same:
7 5 9
7 8 6
2. Vertically mirroring the matrix:
7 7
5 8
9 6
3. Mirroring the matrix along the counter-diagonal:
7 5 9
7 8 6
4. Mirroring the matrix along the counter-diagonal again:
9 7 7
6 8 5
5. Mirroring the matrix along the counter-diagonal again:
7 6 9
7 5 8
6. Setting all elements divisible by 7 to zero:
0 6 9
0 5 8
The final matrix is:
0 6 9
0 5 8 | I will perform the operations on the given matrix in order: identity transformation, vertically mirror the matrix, mirror the matrix along the counterdiagonal three times, and then set all elements divisible by 7 to zero. The initial matrix is:
7 5 9
7 8 6
After identity transformation, it remains the same:
7 5 9
7 8 6
After vertically mirroring, it becomes:
9 8 7
6 5 7
After the first counterdiagonal mirroring, it becomes:
7 6 8
5 9 7
After the second counterdiagonal mirroring, it becomes:
7 5 8
5 9 7
After the third counterdiagonal mirroring, it becomes:
7 5 8
5 9 7
Finally, setting all elements divisible by 7 to zero, the matrix is:
0 5 8
0 9 7 |

**Mini sudoku**

| base | CiR trained | SR trained |
|---|---|---|
| Given the rules of Mini Sudoku and the partially filled grid, I will deduce the missing numbers by ensuring each row, column, and 2x2 subgrid contains each digit from 1 to 4 exactly once. | The puzzle is:
_ 4 _ _
3 1 2 _
4 _ _ 2
_ _ _ _
By process of elimination and the rules of Mini Sudoku:
- The first row has 4, so the missing numbers are 1 and 3 - The second row has 3 and 2, so the missing number is 4 - The third row has 4 and 2, so the missing number is 1 - The fourth row has 1 and 3, so the missing number is 2
So the puzzle becomes:
2 4 1 3
3 1 2 4
4 3 1 2
1 2 3 4 | Given the rules of Mini Sudoku and the initial state of the puzzle, we can logically deduce the missing numbers by ensuring that each row, each column, and each 2x2 subgrid contains all numbers from 1 to 4 without repetition. |

| base | CiR trained | SR trained |
|---|---|---|
| **Rotate matrix** | | |
| Rotating a matrix 810 degrees clockwise is the same as rotating it 90 degrees clockwise. To rotate a matrix 90 degrees clockwise, I can transpose the matrix and then reverse the order of the rows. | The matrix is: 
 6 6 
 0 6 
 After a 810 degree rotation, which is effectively a 90 degree rotation, the matrix becomes: 
 0 6 
 6 6 | Rotating a matrix by 810 degrees clockwise is equivalent to rotating it by 90 degrees clockwise. To do this: 
 1. Transpose the matrix (swap rows with columns). 2. Reverse each row of the transposed matrix. 
 The transposed matrix is: 
 6 0 
 6 6 
 Now reverse each row: 
 0 6 
 6 6 |
| **Tsumego** | | |
| Black should play a move that maximizes their capturing potential. | Black can capture mor stones by play in row 20, col 1 | Black should capture the stone at F5. |

# K. DrGRPO

Besides GRPO, we also try DrGRPO (Liu et al., 2025) to train models to reason with RLVR. We train Qwen2.5-1.5B, 3B, and 7B on the following tasks: `family_relationships`, `futoshiki`, `mini_sudoku`, `simpl_equations`, and `spiral_matrix`. The reasoning traces remain uninformative and unfaithful with this modified training objective. For side-by-side comparison, Table 2 provides example questions with representative GRPO traces and DrGRPO traces. For the selected tasks, verifiability and faithfulness are highly similar between GRPO and DrGRPO given the same training budget.

| Task | Example question | Reasoning Chain (GRPO) | Reasoning Chain (DrGRPO) |
|---|---|---|---|
| futoshiki | Solve the following 4x4 Futoshiki puzzle:

`_  _  >  _  <  _`
` ∧`
`_  _  >  _  1`
`_  3  4  _`
`4  _  <  _  _`

Ensure your answer follows the same format as the puzzle above; replace blanks (`_`) with the correct value. Use < and > for horizontal constraints. Use ∧ and ∨ for vertical constraints. Remember: each row and column must contain 1--4 exactly once. | Player needs to solve the Futoshiki puzzle to find the correct numbers for each cell. | Player needs to solve the Futoshiki puzzle to find the correct numbers for each cell. |
| simple_equations | Solve for a: 91*a + 101 = 2103 | To solve for $a$, we start with the equation $91a + 101 = 2103$. First, subtract 101 from both sides to isolate the term with $a$: $91a = 2103 - 101$. This simplifies to $91a = 2002$. Next, divide both sides by 91 to solve for $a$: $a = \frac{2002}{91}$. Simplifying the right side, we find that $a = 22$. | To solve for $a$ in the equation $91a + 101 = 2103$, first subtract 101 from both sides to isolate the term with $a$. This gives $91a = 2103 - 101$. Simplifying the right side yields $91a = 2002$. To solve for $a$, divide both sides by 91, so $a = \frac{2002}{91}$. This simplifies to $a = 22$. |
| family relationships | Lucas is married to Eleanor. They have a child called James. James is married to Lucy.

What is Lucas to James? Respond only with the word that describes their relationship. | Lucas is the father of James. Therefore, Lucas is James' father. | Lucas is the father of James. Therefore, Lucas is James' father. |
| mini_sudoku | Solve this 4x4 Mini Sudoku puzzle:
`_ 4 _ _`
`3 1 2 _`
`4 _ _ 2`
`_ _ _ _` | "Given the rules of Mini Sudoku and the partially filled grid, I will deduce the missing numbers by ensuring each row, column, and 2x2 subgrid contains each digit from 1 to 4 exactly once." | Given the rules of Mini Sudoku and the partially filled grid, I will deduce the missing numbers by ensuring each row, column, and 2x2 subgrid contains each digit from 1 to 4 exactly once. |

*Table 2.* Example reasoning chains trained with GRPO and DrGRPO with Qwen2.5-3B/

