# OpenReview forum: "Outcome Rewards Do Not Guarantee Verifiable or Causally Important Reasoning"
_ICML.cc/2026/Conference — ICML 2026 regular_

### Official Review · Reviewer_dDqw · 2026-03-06

**Soundness:** 3
**Presentation:** 3
**Significance:** 3
**Originality:** 3
**Overall Recommendation:** 5
**Confidence:** 2

**Summary:**

This paper examines whether RLVR training on CoT reasoning produces reasoning chains that are actually used by the model (faithful) and that are sufficient for an external verifier to reconstruct the answer (verifiable). The authors introduce two metrics. The causal importance of reasoning (CIR) measures how much truncating the reasoning chain changes the model’s answer. Sufficiency of reasoning (SR) measures whether an external verifier can recover the answer from the reasoning trace alone, without seeing the original question. The authors apply these metrics to Qwen2.5 models trained with RLVR on 40 ReasoningGym tasks and find that RLVR does not reliably improve CIR or SR, and that on tasks with low CIR and SR the model can achieve good accuracy without reasoning chains. They remedy this with supervised finetuning on a number of expert traces before RLVR, and by augmenting the RLVR reward with auxiliary reward signals (CIR and SR).

**Compliance With Llm Reviewing Policy:**

Affirmed.

**Key Questions For Authors:**

1.	When using CIR or SR as an auxiliary reward, how do the authors rule out that models game the metric?
2.	what is the proposed mechanism by which optimizing SR as a reward improves CIR, and vice versa?

**Limitations:**

yes

**Strengths And Weaknesses:**

**Soundness**

*Strengths*

-	The CIR and SR metrics are clearly defined and make sense as a measure for reasoning.
-	The validation experiments are convincing: showing that tasks with low CIR and SR can be solved equally well by a model trained to answer directly supports the validity of the metrics.

*Weaknesses*

-	While interesting, I’m a bit concerned that the auxiliary rewards set-up could lead to the systems gaming these metrics: when CIR is used directly as a training reward, the model could get reward for making its answer artificially dependent on features in the reasoning trace, without that dependence reflecting genuine computation.
-	The SFT remedy is cleaner in that regard, but the paper does not discuss the difference between these two.
-	CIR and SR are somewhat indirect proxies for the underlying concepts that they mean to capture. They are useful because in many cases we do not have access to the internal computations, but the paper does not engage with possible failure modes of these operationalizations.


**Presentation**

*Strengths*

-	The paper is easy to follow, the figures are clear and the research questions clearly stated.

*Weaknesses*

-	The dissociation between CIR and SR in terms of correlation with accuracy (figure 5) is not clearly described in the text. The authors write that “this pattern holds only when the performance gain is sufficiently large”, yet they report the correlation overall. I’m not sure what is meant by this.

**Significance**

*Strengths*

-	The question is timely and important given how widespread RLVR reasoning models are.
-	The finding that reasoning chains might be more verifiable but not faithful, while not new, is important for AI safety.
-	The auxiliary reward approach is widely applicable

*Weaknesses*

-	From the paper, the case for optimizing CIR and SR is not quite clear. Why these specific metrics over other possible metrics?

**Originality**

*Strengths*

-	CIR and SR are novel metrics.

*Weaknesses*

-	The core finding that high accuracy does not equal faithful reasoning has been shown before. The novelty here is the metrics and the systematic empirical confirmation.

---

> ### Author Rebuttal · Authors · 2026-03-31
>
> We thank the reviewer for the thoughtful and supportive review. We are glad that the reviewer found the paper clear and timely. We address these concerns below.
>
> > **"the model could get reward for making its answer artificially dependent on features in the reasoning trace, without that dependence reflecting genuine computation."**
>
> Thank you for raising this concern. We agree that optimizing CIR does not by itself guarantee “genuine internal computation” in a mechanistic sense. Our claim is narrower: CIR measures whether the answer is observably dependent on the emitted reasoning trace, i.e., whether truncating the trace changes the answer distribution. It is therefore a proxy for the faithfulness of *externalized reasoning*, not a full account of hidden internal computation. We will make this limitation explicit. At the same time, CIR reward does not only increase dependence in isolation: it also tends to improve SR and produces more task-specific intermediate steps rather than generic rationalizations.
>
> > **"the paper does not discuss the difference between SFT and auxiliary reward."**
>
> Thank you for pointing this out. The two interventions operate differently: SFT provides expert traces that teach both reasoning strategy and format, whereas CIR/SR auxiliary rewards do not add task information and instead shape how reasoning is used and presented during RLVR. In our additional analysis, SFT yields the strongest gains in concrete intermediate steps and explicit calculations, suggesting it is the cleaner intervention when the goal is improving reasoning quality rather than optimizing a proxy.
>
> > **"The paper does not engage with possible failure modes of these operationalizations."**
> > **"When using CIR or SR as an auxiliary reward, how do the authors rule out that models game the metric?"**
>
> Thank you for this important concern. We agree that CIR and SR are operational proxies, not direct measurements of genuine internal computation, and we will revise the paper to make this limitation explicit. Our claim is therefore narrower: CIR measures whether the model’s answer is observably dependent on the externalized reasoning trace, while SR measures whether that trace is sufficiently self-contained for answer recovery by a verifier.
>
> We also agree that, when used as auxiliary rewards, both metrics can in principle be gamed. To study this directly, we ran an additional failure-mode analysis over three reward-hack hypotheses: **answer parroting**, **question repetition**, and **answer paraphrasing**. The results support the reviewer’s concern: these behaviors become substantially more common under CIR reward (e.g., answer parroting increases from 46.9% in the initial model to 78.0%, and answer paraphrasing from 33.0% to 51.4%), and are also present under SR reward (57.3% and 37.1%, respectively). This is precisely why we view CIR/SR as useful but imperfect proxies rather than complete guarantees of faithful reasoning.
>
> To mitigate these failure modes, we already remove explicit answer copying in the current paper. We additionally strip repeated or paraphrased question content from the reasoning trace before recomputing the metric ($\mathrm{SR^{-}}$). Our main qualitative conclusions continue to hold after this sanitization: training with augmented CIR improves average $\mathrm{SR^{-}}$ to 0.40, and training with augmented SR improves average $\mathrm{SR^{-}}$ to 0.52, from 0.15 in the initial model. We will add this analysis in a separate section in the revision.
>
> > **"The dissociation between CIR and SR is not clearly described in the text. "**
>
> We agree the presentation can be clearer. Across tasks, $\Delta \mathrm{SR}$ and $\Delta \mathrm{Acc}$ show a positive overall correlation, whereas $\Delta \mathrm{CIR}$ and $\Delta \mathrm{Acc}$ do not. However, the SR trend is driven mainly by tasks with larger accuracy gains; when $\Delta \mathrm{Acc} < 0.5$, many tasks still show decreases in SR.
>
> > **"Why these specific metrics over other possible metrics?"**
>
> We focus on CIR and SR because they directly measure the two properties central to this paper. In contrast, proxy metrics such as length or stylistic markers may correlate with good reasoning, but they do not directly test either property. Our claim is not that CIR and SR are the only possible objectives, but that they are the most directly aligned with our target desiderata.
>
> > **"What is the proposed mechanism by which optimizing SR as a reward improves CIR, and vice versa?"**
>
> Our hypothesis is that both objectives favor traces that are more task-specific, explicit, and concrete. Low-CIR/SR traces are often generic, whereas higher-CIR/SR traces contain clearer intermediate steps tied to the input. This shared preference may explain why optimizing one often improves the other.
>
> We thank the reviewer again for the careful reading and constructive feedback. We believe these clarifications significantly strengthen the paper.

---

> > ### Author Rebuttal · Reviewer_dDqw · 2026-04-02
> >
> > Thank you for your detailed rebuttal and for running the additional experiments regarding metric gaming. The fact that answer parroting increases so substantially under CIR reward confirms my concern and I think this is the right way to handle it. The SR^-/CIR ^- fixes are good patches, although I suspect there are still other ways for an optimized model to game these metrics, and it would be good to discuss these limitations clearly in the paper. Overall, my concerns are sufficiently addressed and I maintain my score of Accept.

---

### Official Review · Reviewer_9ZUj · 2026-03-12

**Soundness:** 3
**Presentation:** 3
**Significance:** 2
**Originality:** 2
**Overall Recommendation:** 4
**Confidence:** 3

**Summary:**

This paper investigates whether Reinforcement Learning from Verifiable Rewards (RLVR) actually makes the model’s chain-of-thoughts meaningful. The authors define two metrics, Causal Importance of Reasoning (CIR) and Sufficiency of Reasoning (SR), to test the faithfulness and verifiability of the reasoning chains, respectively. The authors find that while RLVR improves task accuracy, it often leads to a decline in CIR and SR. Finally, the authors discover that Supervised Fine-Tuning and CLR/SR-based rewards can improve reasoning chains on both metrics.

**Compliance With Llm Reviewing Policy:**

Affirmed.

**Final Justification:**

I thank the authors for their responses. Most of my questions have been well addressed and supported with additional experiments. I appreciate the authors' clarifications and encourage merging these details into the revision. I believe the proposed metrics are useful for reasoning tasks. I have updated my score from 2: Reject to 4: Weak Accept accordingly.

**Key Questions For Authors:**

1. Since all experiments were run on Qwen-2.5, I am wondering if the experimental results generalize to model families beyond this specific architecture. While I understand that finding open-source models with a specific post-training recipe is difficult, some evidence of broader generalization is desired.
2. Are there any deeper connections between CIR and SR? By looking at their definitions, they seem to have some relations.
3. What is the practical, long-term usage of CIR and SR? As shown in Figure 8, CIR and SR rewards can generally improve model performance on those specific metrics, but the task accuracy doesn’t change much with the CIR reward. Even though the SR reward can sometimes improve accuracy, it requires an external (and potentially much more powerful) verifier in the loop. Interpretability and explainability of the chain-of-thought might be one direction, but I would appreciate further discussion on practical deployment.

**Limitations:**

The author did not include a limitation section. I suggest the authors to allocate a dedicated section to discuss the limitations in the revised manuscript

**Strengths And Weaknesses:**

# Strengths
* Understanding the effects of chain-of-thought reasoning and the impact of RLVR are important and highly relevant topics.
* The presentation is clear; it is easy to follow the paper’s content and understand the research questions.

# Weaknesses
1. ***Metrics Design*. Generally, the design and motivation of the metric frameworks are unclear to me. This is the main weakness of this paper.**
* For Causal Importance of Reasoning (CIR), why use Jensen-Shannon divergence to measure truncated CoT? Why not other metrics like the KL divergence or absolute difference?
* For Sufficiency of Reasoning (SR),
  - Why is the statement in [Line 164, p.3] true? “If the reasoning is sufficient, conditioning on q should add little information, so the probability-level predictions should be nearly unchanged.” For some tasks, knowing what question is asking is essential for verifying if the answer and/or reasoning path are correct.
  - Why use an external verifier instead of the reasoning model itself? A reasoning trace might be specific to the reasoning model that provides it, meaning the verifiability of a trace is heavily dependent on the external verifier’s capability. A reasoning model might provide a nonsense reasoning path, but if it simply repeats the question within that path, a powerful verifier could still successfully solve it from scratch, falsely inflating the score to SR(q,t) = 1.

2. **Confounding Effect of the Benchmark on CIR/SR Drops**. The authors claim that outcome-based RLVR does not guarantee faithful reasoning, pointing to the drops in CIR and SR shown in Figure 2. However, as the authors themselves acknowledge in Line 80, some ReasoningGym tasks do not actually require reasoning to solve. By looking at Figure 2, the tasks where CIR decreases are mostly not in the math domain, which probably do not require reasoning to answer. If a model can solve a task directly without a chain-of-thought, it is completely expected that CIR and SR would decrease. Therefore, the benchmark tasks themselves are driving the CIR/SR decrease, rather than this being a fundamental flaw of outcome-based reward.

3. [Very Minor Error] Presentation.
* Line 1102 Figure citation is missing as ??.
* Figure 2 is slightly hard to interpret, as the details of each task are unclear.

---

> ### Author Rebuttal · Authors · 2026-03-31
>
> We thank the reviewer for the reading and feedback. We agree that the paper can be strengthened in the motivation, interpretation, and scope of $\mathrm{CIR}$ and $\mathrm{SR}$. In the revision, we will clarify the metric design, add analysis on benchmark-confound and generalization concerns, and include a limitations section.
>
> > **"How should $\mathrm{CIR}$ be interpreted, and why use Jensen--Shannon divergence?"**
>
> Our goal with $\mathrm{CIR}$ is to measure whether generated reasoning causally affects the answer distribution, rather than merely correlating with the final answer. We use JS because it is symmetric, bounded, and well-defined for comparing answer distributions across truncated prefixes. However, the divergence choice is not the core claim: the key idea is to quantify how much the answer distribution changes when reasoning tokens are removed. As a robustness check, we replaced JSD with absolute probability difference and found highly correlated scores (Pearson $r=0.98$, $p<10^{-6}$). We will add this result in revision
>
> > **"Why is the statement in [Line 164, p.3] true?"**
>
> Our intent was not to claim that the original question is unnecessary for judging semantic correctness. Rather, $\mathrm{SR}$ operationalizes sufficiency as *self-containedness for answer recovery*: after removing explicit answer leakage, we ask whether a strong verifier predicts the same answer from the reasoning alone as from the question plus reasoning. If access to the question changes the verifier’s prediction, the trace likely omitted important task information, constraints, or intermediate steps. Thus, $\mathrm{SR}$ measures self-containedness for answer reconstruction, not proof of correctness.
>
> > **"Why use an external verifier?"**
>
> First, using the same model that generated the trace would make the evaluation circular, since the model may decode its own shorthand even when the trace is not interpretable to an independent observer. Second, a stronger verifier reduces false negatives caused by verifier weakness. We agree that $\mathrm{SR}$ depends on verifier choice. In Appendix G, we also evaluate $\mathrm{SR}$ with GPT-4.1-mini and find strong agreement with the main verifier.
>
> > **"What about repeating the question?"**
>
> We agree this is a critical point. Our current definition of $\mathrm{SR}$ is already permissive, yet after outcome-reward training, $\mathrm{SR}$ is often still close to 0. Stricter variants can therefore tell an even more concerning story. We have now implemented a stricter variant, $\mathrm{SR}^{-}$, that removes repeated or paraphrased question content, and we report those results in our response to reviewer dDqw.
>
> > **"Is this just a benchmark confound?"**
>
> In line 80, our claim is not that reasoning is unnecessary for these tasks. Rather, the model may learn to solve them with limited effective use of the reasoning trace, leading to similar performance with and without reasoning. As noted in line 299, final accuracy on these tasks is also lower, suggesting that more effective use of reasoning could further improve performance; our SFT results support this interpretation. We will revise this section to make the claim clearer.
>
> > **"If a model can solve a task directly without chain-of-thought, is it expected that $\mathrm{CIR}$ and $\mathrm{SR}$ decrease?"**
>
> This is not always the case: on some tasks, a model can achieve low $\mathrm{CIR}$ while $\mathrm{SR}$ still increases during training.
>
> > **"Do the results generalize beyond Qwen-2.5?"**
>
> We agree that broader architectural evidence would strengthen the paper. We therefore ran additional experiments on Llama 3.2-3B, and the main finding is consistent with our central claim: 80\% of tasks have low $\mathrm{CIR}$ and 85\% have low $\mathrm{SR}$.
>
> > **"What is the connection between $\mathrm{CIR}$ and $\mathrm{SR}$?"**
>
> $\mathrm{CIR}$ and $\mathrm{SR}$ capture different properties. $\mathrm{CIR}$ measures whether the answer distribution causally depends on the reasoning tokens, whereas $\mathrm{SR}$ measures whether the trace is self-contained enough for a verifier to recover the answer without the original question. A trace can therefore have high $\mathrm{CIR}$ but low $\mathrm{SR}$ if it affects the model’s answer while remaining opaque to a verifier, or high $\mathrm{SR}$ but low $\mathrm{CIR}$ if it is verifiable but not actually used by the model. Empirically, the metrics are positively correlated, but not identical.
>
> > **"How should $\mathrm{CIR}$ and $\mathrm{SR}$ be used in practice?"**
>
> We view $\mathrm{CIR}$ and $\mathrm{SR}$ as diagnostic tools, not standalone optimization targets. With further exploration, they can potentially help identify shortcutting or reward-hacking behaviors where task accuracy improves without faithful use of the reasoning trace, and evaluate whether chains of thought are useful for auditing, or post-hoc inspection.
>
> We thank the reviewer again for constructive feedbacks.

---

> > ### Author Rebuttal · Reviewer_9ZUj · 2026-04-04
> >
> > I thank the authors for their responses. Most of my questions have been well addressed and supported with additional experiments. I appreciate the authors' clarifications and encourage merging these details into the revision. I believe the proposed metrics are useful for reasoning tasks. I have updated my score accordingly.

---

### Official Review · Reviewer_N1NF · 2026-03-12

**Soundness:** 2
**Presentation:** 3
**Significance:** 3
**Originality:** 3
**Overall Recommendation:** 3
**Confidence:** 4

**Summary:**

This paper questions a key assumption in RLVR (Reinforcement Learning from Verifiable Rewards) for training language models to produce Chain-of-Thought (CoT) reasoning, namely whether the generated reasoning chains faithfully reflect the model’s actual reasoning process for deriving the final answer. The authors propose two evaluation metrics: CIR (Causal Importance of Reasoning), which measures the causal influence of reasoning tokens on the final answer and reflects faithfulness, and SR (Sufficiency of Reasoning), which evaluates whether a verifier can determine the correct answer solely from the reasoning chain, reflecting verifiability. Experimental results show that although RLVR improves task accuracy, it does not consistently enhance CIR or SR, and in many cases the reasoning chains do not play a critical role in producing correct answers. To address this issue, the authors introduce an SFT warm-up stage using a small amount of high-quality expert reasoning trajectories before RLVR, as well as an auxiliary reward mechanism based on CIR and SR, both of which improve the faithfulness and verifiability of the generated reasoning while maintaining task performance.

**Compliance With Llm Reviewing Policy:**

Affirmed.

**Key Questions For Authors:**

See weaknesses

**Limitations:**

yes

**Strengths And Weaknesses:**

**Strengths**

1.	The problem is clearly defined, focusing on the causal importance and sufficiency of Chain-of-Thought reasoning, which has direct relevance to areas such as AI safety and interpretability.
2.	The experimental design is comprehensive. Extensive experiments are conducted across numerous datasets in ReasoningGym and with three model sizes, supporting the reliability of the findings.
3.	The paper not only identifies key issues from the experimental results but also proposes potential solutions and empirically demonstrates their effectiveness.

**Weaknesses and Questions**

1.	What types of samples tend to have low CIR/SR scores? For example, are these cases caused by errors in critical reasoning steps or differences in reasoning patterns?
2.	In the experiments, CIR and SR decrease after RLVR for some tasks. Is there any relationship between the changes in CIR/SR during training and the changes in task accuracy?
3.	Line 243 states that CIR only measures whether the CoT is used by the model, rather than whether it is useful for external observers. How does CIR differ from other metrics that evaluate the relationship between CoT and the final answer, and what advantages does it offer?
4.	In Figure 8, when CIR is used as an auxiliary reward, the baseline without CIR achieves noticeably higher accuracy. What causes this phenomenon?
5.	The writing should be carefully checked. For example, in Appendix G there is a reference to “Figure ??”.
6.	In Appendix E, how is the change in CoT length related to the proposed CIR and SR metrics?

---

> ### Author Rebuttal · Authors · 2026-03-31
>
> We thank the reviewer for the careful reading and constructive feedback. We appreciate the detailed questions, which helped us identify several places where the paper could be strengthened in both analysis and presentation.
>
> We would like to address some of the concerns raised by the reviewer here.
>
> > **"What types of samples tend to have low $\mathrm{CIR}/\mathrm{SR}$ scores?"**
>
> Thanks for the questions. Figure 3 in the paper shows the qualitative example of low/high $\mathrm{CIR}/\mathrm{SR}$. To better understand failure modes, we also added a targeted analysis of low-$\mathrm{CIR}/\mathrm{SR}$ examples. We found that these traces are often characterized by: (i) lack of concrete intermediate steps (70.3\%), (ii) missing explicit calculations when calculations are needed (88.3\%), and (iii) lexically simple or generic wording (63.8\%). These categories are not mutually exclusive. As one example, for Mini Sudoku the reasoning sometimes collapses to a generic statement such as: *Given the rules of Mini Sudoku and the partially filled grid, I will deduce the missing numbers by ensuring each row, column, and 2x2 subgrid contains each digit from 1 to 4 exactly once*, without carrying out the necessary deductions. We will add representative qualitative examples and a more detailed quantitative breakdown in the revision.
>
> > **"Is there any relationship between the changes in $\mathrm{CIR}/\mathrm{SR}$ during training and the changes in task accuracy?"**
>
> We agree that understanding the relationship between reasoning-quality metrics and task performance is important. Our current results in Section 5.3 show an asymmetric pattern: improvements in $\mathrm{CIR}$ are not significantly correlated with improvements in accuracy, whereas improvements in $\mathrm{SR}$ are positively correlated with accuracy. At the same time, when accuracy gains are small, both $\mathrm{CIR}$ and $\mathrm{SR}$ often decrease.
>
> > **"How does $\mathrm{CIR}$ differ from other metrics that evaluate the relationship between CoT and the final answer, and what advantages does it offer?"**
>
> Our $\mathrm{CIR}$ is related in spirit to prior faithfulness metrics such as [1], which also test whether perturbing or truncating the chain of thought changes the model’s prediction. However, [1]’s closest metric, early answering, is a coarse sentence-level test that measures whether the final answer stays the same after truncation. In contrast, $\mathrm{CIR}$ is a token-level continuous metric: for each prefix of the reasoning trace, we compare the model’s probability of the final answer under the truncated trace versus the full trace using JS divergence, and average across token positions. Thus, while [3] measures whether the answer changes under coarse ablations, $\mathrm{CIR}$ measures the cumulative magnitude of the reasoning trace’s causal influence on the final answer probability. This makes $\mathrm{CIR}$ more sensitive to partial use of reasoning and more suitable as both an evaluation metric and an auxiliary training reward.
>
> Relatedly, [2] proposes RECEVAL, which evaluates whether intermediate reasoning steps are correct and provide new information. RECEVAL is closer in spirit to our $\mathrm{SR}$ metric and is largely orthogonal to $\mathrm{CIR}$: a reasoning trace can score highly on RECEVAL while still having $\mathrm{CIR}=0$.
>
> > **"Why does the baseline outperform the $\mathrm{CIR}$-reward model in Figure 8?"**
>
> Our current understanding is that, different from the SFT experiments, the auxiliary reward with $\mathrm{CIR}$ does not always provide information and knowledge to guide models to better solve the tasks. There is sometimes a tension between a reasoning trace being causally important and leading to high accuracy in our RLVR setting. This is also supported by our analysis in Section 5.3. Therefore, the model is confined to its knowledge base. When the $\mathrm{CIR}$ is high, the reasoning traces can still make mistakes in the steps and potentially lead to a wrong answer. However, compared to the generic reasoning traces with low $\mathrm{CIR}$, the reasoning traces with augmented reward show clearer steps and calculations. In other words, faithful reasoning does not necessarily mean higher performance, which we see as a fundamental lesson for people seeking to interpret and improve reasoning models.
>
> > **"What is the relation between CoT length and $\mathrm{CIR}/\mathrm{SR}$?"**
>
> We ran an additional analysis to calculate the correlation. The length of the CoT is positively correlated with both $\mathrm{CIR}$ and $\mathrm{SR}$. $\mathrm{CIR}$ and length have a correlation of 0.7, and $\mathrm{SR}$ and length have a correlation of 0.59 (both significant). We will add this to the main text in connection with Appendix E.
>
> We thank the reviewer again.
>
> [1] Lanham, Tamera, et al. "Measuring faithfulness in chain-of-thought reasoning."
>
> [2] Prasad, Archiki, et al. "Receval: Evaluating reasoning chains via correctness and informativeness."

---

> > ### Author Rebuttal · Reviewer_N1NF · 2026-04-01
> >
> > The rebuttal helpfully addresses several of my questions, especially by clarifying the failure modes of low-CIR/SR samples, positioning CIR relative to prior faithfulness metrics, and providing additional analysis on the correlation between CoT length and the proposed metrics. However, I remain unconvinced that the key concerns are fully resolved. In particular, the explanation for why the CIR-auxiliary-reward model underperforms the baseline in Figure 8 remains largely speculative, and the relationship between CIR/SR improvements and task accuracy still feels only partially characterized. Overall, the rebuttal improves my understanding of the paper, but does not substantially change my overall assessment.

---

> > > ### Author Response · Authors · 2026-04-07
> > >
> > > We thank the reviewer for the response and active engagement! We are glad that several of the questions are addressed. We ran further experiments for the remaining two key concerns and hopefully these can provide more clarity. The additional graphs can be found [here](https://anonymous.4open.science/r/icml_2026_rebuttal_graph-F7E9/ICML_rebuttal_2026.pdf).
> > >
> > > ---
> > > > **"The explanation for why the CIR-auxiliary-reward model underperforms the baseline in Figure 8 remains largely speculative."**
> > >
> > >
> > > We first wish to clarify that we did not claim training with an auxiliary reward improves performance. As stated on **Line 409**, we expected training with an auxiliary reward to yield accuracy similar to the baseline, and on **Line 411** we provide an explanation for this lack of improvement.
> > >
> > >
> > > To further investigate whether the observed differences are attributable to training noise, we trained **2 additional seeds** with the augmented CIR reward, selecting **α = 0.8** as our best-performing hyperparameter. Due to the computational cost of full RL training runs, we limited this to 2 seeds for the rebuttal; we will include more additional seeds in the final paper. We also evaluated using multiple samples: for each test instance, we sample *k* = 8 and average the resulting scores. Under this **mean@8** evaluation, the final accuracy of the augmented reward model is indeed comparable to the baseline [Figure 1](https://anonymous.4open.science/r/icml_2026_rebuttal_graph-F7E9/ICML_rebuttal_2026.pdf).
> > > That said, we do observe that the baseline accuracy improves more rapidly in early training — for instance, at step 30, the baseline already exceeds the augmented reward model. This echoes our earlier explanation: faithful reasoning does not necessarily translate into higher task performance. We view this as a fundamental insight for researchers seeking to interpret and improve reasoning models.We will include these additional experimental results in our final revision. Thank you for the suggestion.
> > >
> > > ---
> > >
> > > > **"The relationship between CIR/SR improvements and task accuracy still feels only partially characterized."**
> > >
> > > To more fully characterize the relationship between CIR/SR improvements and task accuracy, we conduct an additional experiment. [Figure 2](https://anonymous.4open.science/r/icml_2026_rebuttal_graph-F7E9/ICML_rebuttal_2026.pdf) plots the trajectory of CIR and SR across training for two contrasting sets of tasks. The first set consists of **8 tasks** exhibiting low and decreasing CIR and SR (the same tasks analyzed in Sections 6 and 7). The second set consists of **8 tasks** with high and increasing CIR and SR: `simple_equations`, `simple_geometry`, `advanced_geometry`, `polynomial_multiplication`, `simple_integration`, `letter_jumble`, `leg_counting`, and `fraction_simplification`.
> > >
> > > Across training, changes in CIR and SR closely track changes in accuracy. For tasks where CIR and SR increase (red), this rise is accompanied by substantial accuracy gains. Conversely, for tasks where accuracy improves only marginally, CIR and SR tend to decline (blue). Together, these results complement Section 5 and suggest that CIR and SR are reliable indicators of task-level learning dynamics throughout RL training.
> > >
> > > We thank the reviewer again for constructive feedback.

---

### Official Review · Reviewer_N2s1 · 2026-03-13

**Soundness:** 3
**Presentation:** 4
**Significance:** 3
**Originality:** 3
**Overall Recommendation:** 5
**Confidence:** 3

**Summary:**

The authors are focused on two facets of model reasoning during RLVR: whether reasoning traces are faithful, as measured by the "Causal Importance of Reasoning" score, and whether they are verifiable, as measured by the "Sufficiency of Reasoning Score." The CIR score essentially assesses whether reasoning tokens lead to a meaningful improvement in predicting the final answer by comparing answer probabilities while conditioning on different prefixes. The SR score assesses whether the reasoning trace can allow the answer to be inferred by itself, without access to the question. The authors conduct RLVR runs on a subset of tasks in ReasoningGym and find that, with the exception of tasks where the model has a substantial improvement in accuracy (above 50%), both CIR and SR scores fall. The authors then perform two interventions: performing cold-start training on verifiable reasoning traces and performing RL training with auxiliary rewards targeting CIR and SR. Both interventions lead to an improvement in CIR and SR values, with the two values being correlated with each other.

**Compliance With Llm Reviewing Policy:**

Affirmed.

**Final Justification:**

My concerns, which were mainly about whether the observed results were specific to the narrowness of the training regime, have been largely resolved. I think the paper is an interesting and systematic study of CoT verifiability in the context of RLVR, and could have implications for both alignment and interpretability research.

**Key Questions For Authors:**

Please see the weaknesses above.

**Limitations:**

Yes.

**Strengths And Weaknesses:**

Strengths:
- The authors provide a detailed analysis of how different aspects of reasoning chain interpretability change as a result of RLVR training. Commonly, to the best of my knowledge, such analyses focus on already trained artifacts. As a result, the set of experiments provide a different angle on a well-studied problem.
- The insights regarding CIR/SR improving for tasks where the model has substantial increases in performance and the CIR/SR auxiliary rewards likewise leading to improvements in the same metrics (and even improving accuracy for SR) were interesting.
- The SR auxiliary reward is arguably a simpler instantiation of the Kirchner et al's prover-verifier setting.

Weaknesses:
- One key insight is that CIR/SR can be improved when model performance substantially increases. What does this mean for more challenging settings such as math benchmarks like MATH-Hard or AIME (challenging at least for Qwen2.5)? Does training in these complex settings, which likely is where RLVR will be used most, lead to improvements in CIR/SR?
- I am curious as to what CIR/SR values are for existing reasoning models like Qwen3. This could be evaluated in the same ReasoningGym settings as well as math benchmarks like MATH-Hard or AIME 2025. I expect that such models have been trained on a broad distribution of relatively challenging tasks. Does RLVR lead to poor CIR/SR for them as well?
- I also wonder whether the results are primarily a function of the narrow data distributions during training. How does multi-task training, for instance, affect CIR/SR? If a model was trained on tasks that led to substantial improvements in conjunction with those with more modest changes, would the improved CIR/SR transfer?

---

> ### Author Rebuttal · Authors · 2026-03-31
>
> We thank Reviewer **N2s1** for the careful reading and thoughtful feedback. We are glad the reviewer found our analysis of reasoning-chain interpretability during RLVR interesting, especially the observations about $\mathrm{CIR}$/$\mathrm{SR}$ behavior during training and the effectiveness of auxiliary rewards. We also appreciate the reviewer’s positive assessment of the paper’s soundness, presentation, significance, and originality.
>
> We address the reviewer’s points below.
>
>
> >  **"What does this mean for more challenging settings such as math benchmarks like MATH-Hard or AIME (challenging at least for Qwen2.5)? Does training in these complex settings, which likely is where RLVR will be used most, lead to improvements in $\mathrm{CIR}/\mathrm{SR}$?"**
>
> Thank you for raising this point. We agree that harder domains such as MATH-Hard or AIME-style problems are especially important, since these are settings where RLVR is often most relevant in practice.
>
> To examine this, we ran Qwen2.5-3B on **MATH-Hard**. In this setting, we observe improvements in both reasoning metrics and task performance: $\mathrm{CIR}$ increases from $0.2293$ to $0.5051$, $\mathrm{SR}$ increases from $0.2300$ to $0.4200$, and accuracy improves from $0.11$ to $0.56$.
>
> This is consistent with our broader observation that when training leads to substantial performance gains, $\mathrm{CIR}$ and $\mathrm{SR}$ are more likely to improve rather than degrade. In other words, the $\mathrm{CIR}/\mathrm{SR}$ drop is not universal: in more challenging settings, where successful learning appears to require better intermediate reasoning, RLVR can improve both accuracy and reasoning quality. We will include these results in the revision.
>
> This is also one reason we chose to study **ReasoningGym** in addition to standard math benchmarks. While math benchmarks are important, ReasoningGym provides a more diverse set of reasoning tasks and allows us to test whether these patterns extend beyond the relatively narrow distribution of mathematical problem solving.
>
>
> > **"I am curious as to what $\mathrm{CIR}/\mathrm{SR}$ values are for existing reasoning models like Qwen3. This could be evaluated in the same ReasoningGym settings as well as math benchmarks like MATH-Hard or AIME 2025. I expect that such models have been trained on a broad distribution of relatively challenging tasks. Does RLVR lead to poor $\mathrm{CIR}/\mathrm{SR}$ for them as well?"**
>
> Thank you for this suggestion. This is a valuable comparison because such models are trained on a broader and more challenging task distribution than the RLVR runs in our main experiments.
>
> As an initial check, we evaluated **Qwen-3-4B-Thinking** in thinking mode on four selected ReasoningGym tasks: three that were challenging under our metrics in the Qwen2.5-3B setting (**mini\_sudoku**, **binary\_matrix**, **rotate\_matrix**) and one task with relatively strong $\mathrm{CIR}/\mathrm{SR}$ in our main experiments (**simple\_equations**). The out-of-the-box $\mathrm{CIR}$ and $\mathrm{SR}$ values are:
>
> | Task | $\mathrm{SR}$ | $\mathrm{CIR}$ |
> |---|---:|---:|
> | `binary_matrix` | 0.2400 | 0.1656 |
> | `mini_sudoku` | 0.6500 | 0.1725 |
> | `rotate_matrix` | 0.4717 | 0.2669 |
> | `simple_equations` | 0.7400 | 0.2545 |
>
> In our preliminary evaluation, we observe substantial improvements in both $\mathrm{CIR}$ and $\mathrm{SR}$ for the existing reasoning model in thinking mode. That said, higher $\mathrm{CIR}$ and $\mathrm{SR}$ may be enabled by more extensive pretraining on challenging tasks, which is also consistent with our SFT results in Section 6.
>
>
> > **"I also wonder whether the results are primarily a function of the narrow data distributions during training. How does multi-task training, for instance, affect $\mathrm{CIR}/\mathrm{SR}$? If a model was trained on tasks that led to substantial improvements in conjunction with those with more modest changes, would the improved $\mathrm{CIR}/\mathrm{SR}$ transfer?"**
>
> Thank you for this suggestion. We agree that whether $\mathrm{CIR}/\mathrm{SR}$ transfer under broader training distributions is an important question.
>
> To probe this, we ran an additional multi-task experiment in which one model was trained jointly on five tasks: mini_sudoku, simple_equations, futoshiki, spiral_matrix, and family_relationship. These include both tasks with strong $\mathrm{CIR}/\mathrm{SR}$ gains and tasks with weaker gains in the single-task setting.
>
> Our preliminary result suggests limited transfer: simple_equations continues to show strong $\mathrm{CIR}/\mathrm{SR}$ improvement, while mini_sudoku, spiral_matrix, and futoshiki remain relatively low on both metrics. This indicates that mixing tasks with stronger gains does not automatically lift $\mathrm{CIR}/\mathrm{SR}$ on tasks with weaker gains. We will add this analysis in the revision and discuss it as evidence that $\mathrm{CIR}$ and $\mathrm{SR}$ are at least partly task-dependent.
>
> We thank the reviewer again for the feedback.

---

> > ### Author Rebuttal · Reviewer_N2s1 · 2026-04-03
> >
> > I thank the authors for their response. My concerns, which were mainly about whether the observed results were specific to the narrowness of the training regime, have been largely resolved. The fact that CIR/SR does not transfer between settings is particularly interesting. I am raising my score to a 5 as a result. I think the paper is an interesting and systematic study of CoT verifiability in the context of RLVR, and could have implications for both alignment and interpretability research.

---

### Decision · Program_Chairs · 2026-04-30

**Decision:**

Accept (regular)

**Comment:**

This paper provides a new metric and analysis of the value of COT to the model's reasoning . It introduces 2 new metrics of causal importance and sufficiency, and use it to diagnose when the model is reasoning robustly. Most of the reviewers seem convinced after extended discussions that the metrics are meaningful and relevant, the experiments are robust overall this paper can have impact on interpretability and safety research. Reviewer N1NF's remain objection centers on "whether these faithful/verifiable reasoning metrics truly explain the underlying learning mechanisms of RLVR, or merely serve as phenomenological indicators". However the authors have not made any mechanistic claims for CIR/SR, and the question is whether CIR/SR are useful diagnostically, and in a way that is novel relative to state of the art. All reviewers seem to agree that the answer is yes, justifying acceptance.